# Structural Analysis of Branch-and-Cut and the Learnability of Gomory Mixed Integer Cuts

**Maria-Florina Balcan**
School of Computer Science
Carnegie Mellon University
ninamf@cs.cmu.edu

**Siddharth Prasad**
Computer Science Department
Carnegie Mellon University
sprasad2@cs.cmu.edu

**Tuomas Sandholm**
Computer Science Department
Carnegie Mellon University
Optimized Markets, Inc.
Strategic Machine, Inc.
Strategy Robot, Inc.
sandholm@cs.cmu.edu

**Ellen Vitercik**
Management Science and Engineering Department
Computer Science Department
Stanford University
vitercik@stanford.edu

## Abstract

The incorporation of cutting planes within the branch-and-bound algorithm, known as branch-and-cut, forms the backbone of modern integer programming solvers. These solvers are the foremost method for solving discrete optimization problems and have a vast array of applications in machine learning, operations research, and many other fields. Choosing cutting planes effectively is a major research topic in the theory and practice of integer programming. We conduct a novel structural analysis of branch-and-cut that pins down how every step of the algorithm is affected by changes in the parameters defining the cutting planes added to an integer program. Our main application of this analysis is to derive sample complexity guarantees for using machine learning to determine which cutting planes to apply during branch-and-cut. These guarantees apply to infinite families of cutting planes, such as the family of Gomory mixed integer cuts, which are responsible for the main breakthrough speedups of integer programming solvers. We exploit geometric and combinatorial structure of branch-and-cut in our analysis, which provides a key missing piece for the recent generalization theory of branch-and-cut.

## 1   Introduction

Integer programming (IP) solvers are the most widely-used tools for solving discrete optimization problems. They have many applications in machine learning, operations research, and other fields, including MAP inference [34], combinatorial auctions [52], NLP [37], neural network verification [17], interpretable classification [61], training of optimal decision trees [13], and optimal clustering [48].

Under the hood, IP solvers use the tree-search algorithm branch-and-bound [41] augmented with *cutting planes*, known as *branch-and-cut* (B&C). A cutting plane is a linear constraint that is added to the linear programming (LP) relaxation at any node of the search tree. With a carefully selected cut, the LP guidance can more efficiently lead B&C to the optimal integral solution. Cutting planes, specifically the family of *Gomory mixed integer cuts* (GMI) which we study in this paper, are responsible for breakthrough speedups of modern IP solvers [15, 21].

36th Conference on Neural Information Processing Systems (NeurIPS 2022).

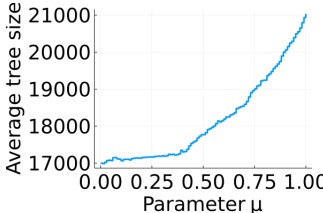 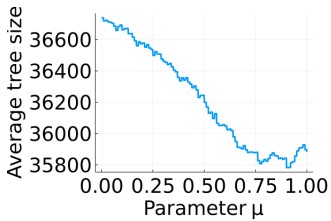

(a) Facility location with 40 locations and 40 clients; sampled by perturbing a base facility location IP.

(b) Facility location with 80 locations, 80 clients, and random Euclidean distance costs.

Figure 1: These figures illustrate the need for distribution-dependent policies for choosing cuts. We plot the average number of nodes B&C expands as a function of a parameter $\mu$ that controls a policy to add GMI cuts, detailed in Appendix A. In each figure, we draw a training set of facility location IPs from two different distributions. In Figure 1a, we define the distribution by starting with a uniformly random facility location instance and perturbing its costs. In Figure 1b, the costs are more structured: the facilities are located along a line and the clients have uniformly random locations. In Figure 1a, a smaller value of $\mu$ leads to small search trees, but in Figure 1b, a larger value of $\mu$ is preferable.

Successfully employing cutting planes can be challenging because there are infinitely many cuts to choose from and there are still many open questions about which cuts to employ when. A growing body of research has studied the use of machine learning for tuning various aspects of IP solvers [e.g., 2, 8, 12, 24, 29, 33, 35, 36, 38, 40, 42, 44, 45, 51, 52, 57–60], recently including cut selection [10, 11, 30, 53, 55]. We analyze a machine learning setting where there is an unknown distribution over IPs—for example, a distribution over a shipping company's routing problems. The learner receives a *training set* of IPs sampled from this distribution which it uses to learn cut parameters with strong average performance over the training set (leading, for example, to small search trees). Figure 1 illustrates that tuning cut parameters according to the instance distribution at hand can have a large impact on B&C's performance, and that for one distribution, the best parameters can be very different—in fact opposite—than the best parameters for another distribution.

We provide *sample complexity bounds* for this procedure, which bound the number of training instances sufficient to ensure that if a set of cut parameters leads to strong average performance over the training set, it will also lead to strong expected performance on future IPs from the same distribution. These guarantees apply *no matter* what procedure is used to optimize the cut parameters over the training set—optimal or suboptimal, automated or manual.

A significant body of research has recently provided sample complexity bounds for automated algorithm configuration, further illustrating the importance of this line of research [e.g., 5–7, 9–11, 16, 25, 28]. However, these works have been unable to analyze Gomory mixed integer (GMI) cuts [26], which are perhaps the most important family of cutting planes in integer programming. They dominate most other families of cutting planes [23] and are directly responsible for the realization that a B&C framework is necessary for the speeds now achievable by modern IP solvers [4]. Prior research has been unable to handle GMI cuts because there are an uncountably infinite number of different GMI cuts that one could add, whereas prior research on cutting planes was only able to handle cutting plane families of finite effective size [10, 11]. The current work closes this gap.

The key challenge is that an infinitesimal change to any GMI cut can completely change the entire course of B&C because a cut added at the root remains in the LP relaxations stored in each node all the way to the leaves. At its core, our analysis therefore involves understanding an intricate interplay between the continuous and discrete components of our problem. The first, continuous component requires us to characterize how an LP's solution changes as a function of its constraints. The optimum will move continuously through space until it jumps from one vertex of the polytope to another. We use this characterization to analyze how the B&C tree—a discrete, combinatorial object—varies as a function of its LP guidance, which allows us to prove our sample complexity bound.

## 1.1 Our contributions

In order to prove our sample complexity bound for GMI cuts, we analyze how the B&C tree varies as a function of the cut parameters on any IP. We prove that the set of all possible cuts can be partitioned

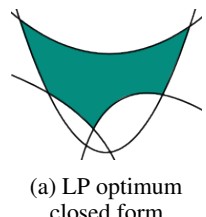 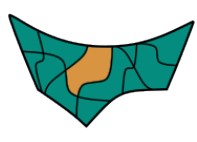 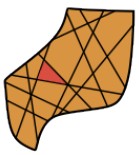 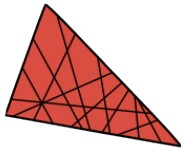

| (a) LP optimum closed form (Lemma 4.1) | (b) Invariant branching (Lemma 4.2) | (c) Invariant LP integrality (Lemma 4.3) | (d) Invariant B&C execution (Theorem 4.4) |

Figure 2: Our B&C analysis involves successive refinements to our partition of the parameter space.

into a finite number of regions such that within any one region, B&C builds the exact same search tree. Moreover, the boundaries between regions are defined by low-degree polynomials. The simplicity of this function allows us to prove our sample complexity bound. The buildup to this result consists of three main contributions, each of which we believe may be of independent interest:

1. Our first main contribution (Section 3) addresses a fundamental question in linear programming: how does an LP's solution change when new constraints are added? As the constraints vary, the solution will jump from vertex to vertex of the LP polytope. We prove that one can partition the set of all possible constraint vectors into a finite number of regions such that within any one region, the LP's solution has a clean closed form. Moreover, we prove that the boundaries defining this partition have a specific form, defined by degree-2 polynomials.

2. We build on this result in our second main contribution (Section 4): a novel analysis of how the entire B&C search tree changes as a function of the cuts added at the root. At a high level, B&C builds this search tree by iteratively subdividing its feasible set via a process called *branching* on variables: in one subdivision, the constraint $x_i \leq k$ is enforced and in the other $x_i \geq k + 1$, for some variable $i$ and integer $k$ ($k$ is chosen according to the LP relaxation solution, as described in Section 2). Upon constructing a subdivision $S$, it checks whether the LP relaxation restricted to $S$ is integral. If not, it further subdivides $S$, unless the LP relaxation's solution is worse than the best integral solution found thus far, in which case it stops searching in $S$—a process called *fathoming* or *pruning $S$*. Each subdivision is stored as a node in B&C's search tree. Our analysis of how the B&C search tree changes as a function of the cuts added has four steps, illustrated by Figure 2:

   (a) Section 4.1: First, we use our result from Section 3 to show that the cut parameter space can be partitioned into regions such that in any one region, the LP optimal solution at any node of the B&C search tree has a clean closed form, as illustrated in Figure 2a.

   (b) Section 4.2: We use this result to show that each region can be further partitioned (as illustrated in Figure 2b) such that no matter what cut we employ in any one region, all of the branching decisions that B&C makes are fixed. Intuitively, this is because the branching decisions depend on the LP relaxation, which has a closed-form solution in any one region.

   (c) Section 4.3: Next, we show that each region from Figure 2b can be further partitioned into regions (illustrated in Figure 2c) where in any one region, for every node in the B&C tree, the integrality of that node's LP relaxation is invariant no matter what cut in that region we use.

   (d) Section 4.4: Finally, we show that each of these regions can be further subdivided into regions (as in Figure 2d) where the nodes that B&C fathoms are fixed, so the tree it builds is fixed.

3. This result allows us to prove sample complexity bounds for learning high-performing cutting planes from the class of GMI cuts, our third main contribution (Section 5). Our key technical insight is that the GMI cutting plane coefficients can be viewed as a mapping that embeds our polynomial partition from the previous step (Figure 2) into the space of GMI cut parameters. We prove that the resulting embedding does not distort the polynomial hypersurfaces too much: the embedded hypersurfaces are still polynomial, with only slightly larger degree.

## 1.2 Related research

**Learning to cut.** Several papers have studied how to use machine learning for cut selection from an applied perspective [30, 53, 55], whereas our goal is to provide theoretical guarantees. Towards this end, this paper helps develop a theory of generalization for cutting plane selection. This line of inquiry began with a paper by Balcan et al. [10], who studied Chvátal-Gomory (CG) cuts [18, 27]

for (pure) integer programs (IPs). Later work [11] provided a unifying sample-complexity analysis of tunable tree-search algorithms when there is a finite set of actions the algorithm can take at any given node. All prior research on generalization guarantees for integer programming [6, 10] fits this framework. In the context of single-variable branching [6], the number of possible branching decisions at any node is equal to the number of variables. In the context of CG cuts, Balcan et al. [10] showed that there are only finitely many distinct cuts at any node. These analyses followed by making pairwise comparisons between actions and understanding in what region of the parameter space one action would be chosen over another. Since there were only a finite number of actions, there were only a finite number of pairwise comparisons. This approach cannot work in our setting due to the uncountably infinite number of GMI cuts. Tackling a continuum of cutting planes requires novel techniques that we develop in this paper—in particular a structural analysis of B&C that is significantly more involved than the finite-action setting.

**Sensitivity analysis of IPs and LPs.** A related line of research studied the *sensitivity* of LPs, and to a lesser extent IPs, to changes in their parameters [e.g., 20, 43, 46]. This paper fits in to this line of research as we study how the solution to an LP varies as new rows are added.

## 2  Notation and branch-and-cut background

**Integer and linear programs.** An *integer program* (IP) is defined by an objective vector $c \in \mathbb{R}^n$, a constraint matrix $A \in \mathbb{Z}^{m \times n}$, and a constraint vector $b \in \mathbb{Z}^m$, with the form

$$\max\{c^T x : A x \leq b, x \geq 0, x \in \mathbb{Z}^n\}. \tag{1}$$

The *linear programming (LP) relaxation* is formed by removing the integrality constraints: $\max\{c^T x : A x \leq b, x \geq 0\}$. We denote the optimal solution to (1) by $x_{\mathsf{IP}}^*$ and its LP-relaxation optimal solution by $x_{\mathsf{LP}}^*$. Let $z_{\mathsf{LP}}^* = c^T x_{\mathsf{LP}}^*$. If $\sigma$ is a set of constraints, we let $x_{\mathsf{IP}}^*(\sigma)$ denote the optimum of (1) subject to these additional constraints (similarly define $z_{\mathsf{LP}}^*(\sigma)$ and $x_{\mathsf{LP}}^*(\sigma)$).

**Polyhedra and polytopes.** A set $\mathcal{P} \subseteq \mathbb{R}^n$ is a *polyhedron* if there exists an integer $m$, $A \in \mathbb{R}^{m \times n}$, and $b \in \mathbb{R}^m$ such that $\mathcal{P} = \{x \in \mathbb{R}^n : A x \leq b\}$. $\mathcal{P}$ is a *rational polyhedron* if there exists $A \in \mathbb{Z}^{m \times n}$ and $b \in \mathbb{Z}^m$ such that $\mathcal{P} = \{x \in \mathbb{R}^n : A x \leq b\}$. A bounded polyhedron is called a *polytope*. The feasible regions of all IPs considered in this paper are assumed to be rational polytopes [1] of full dimension. Let $\mathcal{P} = \{x \in \mathbb{R}^n : a^i x \leq b_i, i \in M\}$ be a nonempty polyhedron. We assume the representation of $\mathcal{P}$ is *irredundant*, that is, $\{x \in \mathbb{R}^n : a^i x \leq b_i, i \in M \setminus \{j\}\} \neq \mathcal{P}$ for all $j \in M$. For any $I \subseteq M$, the set $F_I := \{x \in \mathbb{R}^n : a^i x = b_i, i \in I, a^i x \leq b_i, i \in M \setminus I\}$ is a *face* of $\mathcal{P}$. Conversely, if $F$ is a nonempty face of $\mathcal{P}$, then $F = F_I$ for some $I \subseteq M$. Faces of dimension 1 are called *edges* and faces of dimension 0 are called *vertices*. A detailed reference on the polyhedral theory used in our arguments can be found in Conforti et al. [19].

Given a set of constraints $\sigma$, let $\mathcal{P}(\sigma)$ denote the polyhedron that is the intersection of $\mathcal{P}$ with all inequalities in $\sigma$.

**Cutting planes.** A *cutting plane* is a constraint $\alpha^T x \leq \beta$. Let $\mathcal{P}$ be the feasible region of the LP relaxation of (1) and $\mathcal{P}_{\mathsf{I}} = \mathcal{P} \cap \mathbb{Z}^n$ be the IP's feasible set. A cut is *valid* if it is satisfied by every integer point in $\mathcal{P}_{\mathsf{I}}$: $\alpha^T x \leq \beta$ for all $x \in \mathcal{P}_{\mathsf{I}}$. A valid cut *separates* a point $x \in \mathcal{P} \setminus \mathcal{P}_{\mathsf{I}}$ if $\alpha^T x > \beta$. We refer to a cut both by its parameters $(\alpha, \beta) \in \mathbb{R}^{n+1}$ and the halfspace $\alpha^T x \leq \beta$ in $\mathbb{R}^n$.

An important family of valid cuts that we study in this paper is the set of *Gomory mixed integer (GMI) cuts*. For decades, general-purpose cutting planes were thought to be unwieldy and useless for solving IPs quickly in practice. However, a seminal paper by Balas et al. [4] completely reversed this sentiment by showing that GMI cuts added throughout the B&C tree led to massive speedups. Today, GMI cuts are one of the most important components of state-of-the-art IP solvers.

**Definition 2.1** (Gomory mixed integer cut). Suppose the feasible region of the IP is in equality form $A x = b$, $x \geq 0$ (which can be achieved by adding slack variables). For $u \in \mathbb{R}^m$, let $f_i$

---

[1]This assumption is not a restrictive one. The Minkowski-Weyl theorem states that any polyhedron can be decomposed as the sum of a polytope and its recession cone. All results in this paper can be derived for rational polyhedra by considering the corresponding polytope in the Minkowski-Weyl decomposition.

denote the fractional part of $(\boldsymbol{u}^T A)_i$ and let $f_0$ denote the fractional part of $\boldsymbol{u}^T \boldsymbol{b}$. That is, $(\boldsymbol{u}^T A)_i = (\lfloor \boldsymbol{u}^T A \rfloor)_i + f_i$ and $\boldsymbol{u}^T \boldsymbol{b} = \lfloor \boldsymbol{u}^T \boldsymbol{b} \rfloor + f_0$. The *Gomory mixed integer (GMI) cut* parameterized by $\boldsymbol{u}$ is

$$\sum_{i:f_i \leq f_0} f_i x_i + \frac{f_0}{1 - f_0} \sum_{i:f_i > f_0} (1 - f_i)x_i \geq f_0.$$

The form of the GMI cut is obtained via a slightly more nuanced rounding procedure than the one used to obtain the CG cut $\lfloor \boldsymbol{u}^T A \rfloor \boldsymbol{x} \leq \lfloor \boldsymbol{u}^T \boldsymbol{b} \rfloor$. GMI cuts strictly dominate CG cuts. More details about GMI cuts can be found in the tutorial by Cornuéjols [22].

**Branch-and-cut.** We provide a high-level overview of B&C (Nemhauser and Wolsey [49], for example, provide more details). Given an IP, B&C searches the IP's feasible region by building a binary search tree. B&C solves the LP relaxation of the input IP and then adds any number of cutting planes. It stores this information at the tree's root. Let $\boldsymbol{x}^*_{\mathsf{LP}} = (\boldsymbol{x}^*_{\mathsf{LP}}[1], \ldots, \boldsymbol{x}^*_{\mathsf{LP}}[n])$ be the solution to the LP relaxation with the addition of the cutting planes. B&C next uses a *variable selection policy* to choose a variable $x_i$ to branch on. This means that it splits the IP's feasible region in two: one set where $x_i \leq \lfloor \boldsymbol{x}^*_{\mathsf{LP}}[i] \rfloor$ and the other where $x_i \geq \lceil \boldsymbol{x}^*_{\mathsf{LP}}[i] \rceil$. The left child of the root now corresponds to the IP with a feasible region defined by the first subset and the right child likewise corresponds to the second subset. B&C then chooses a leaf using a *node selection policy* and recurses, adding any number of cutting planes, branching on a variable, and so on. B&C *fathoms* a node—which means that it will never branch on that node—if 1) the LP relaxation at the node is infeasible, 2) the optimal solution to the LP relaxation is integral, or 3) the optimal solution to the LP relaxation is no better than the best integral solution found thus far. Eventually, B&C will fathom every leaf, at which point it has found the globally optimal integral solution. We assume there is a bound $\kappa$ on the size of the tree we allow B&C to build before we terminate, as is common in prior research [6, 10, 11, 31, 38, 39].

Every step of B&C—including node and variable selection and the choice of whether or not to fathom—depends crucially on guidance from LP relaxations. Tighter LP relaxations provide more valuable LP guidance, highlighting the importance of cuts. To give an example, this is true of the *product scoring rule* [1], a popular variable selection policy that our results apply to.

**Definition 2.2.** Let $\boldsymbol{x}^*_{\mathsf{LP}}$ be the solution to the LP relaxation at a node and $z^*_{\mathsf{LP}} = \boldsymbol{c}^T \boldsymbol{x}^*_{\mathsf{LP}}$. The *product scoring rule* branches on the variable $i \in [n]$ that maximizes: $\max\{z^*_{\mathsf{LP}} - z^*_{\mathsf{LP}}(x_i \leq \lfloor \boldsymbol{x}^*_{\mathsf{LP}}[i] \rfloor), 10^{-6}\} \cdot \max\{z^*_{\mathsf{LP}} - z^*_{\mathsf{LP}}(x_i \geq \lceil \boldsymbol{x}^*_{\mathsf{LP}}[i] \rceil), 10^{-6}\}$.

**Polynomial arrangements in Euclidean space.** Let $p \in \mathbb{R}[y_1, \ldots, y_k]$ be a polynomial of degree at most $d$. The polynomial $p$ partitions $\mathbb{R}^k$ into connected components that belong to either $\mathbb{R}^k \setminus \{(y_1, \ldots, y_k) : p(y_1, \ldots, y_k) = 0\}$ or $\{(y_1, \ldots, y_k) : p(y_1, \ldots, y_k) = 0\}$. When we discuss the connected components of $\mathbb{R}^k$ induced by $p$, we include connected components in both these sets. We make this distinction because previous work on sample complexity for data-driven algorithm design oftentimes only needed to consider the connected components of the former set. The number of connected components in both sets is $O(d^k)$ [47, 54, 56].

## 3 Linear programming sensitivity

Our main result in this section addresses a fundamental question in linear programming: how is an LP's optimal solution affected by the addition of new constraints? Later in this paper, we use this result to prove sample complexity bounds for optimizing over the canonical family of GMI cuts.

More formally, fixing an LP with $m$ constraints and $n$ variables, if $\boldsymbol{x}^*_{\mathsf{LP}}(\boldsymbol{\alpha}^T \boldsymbol{x} \leq \beta) \in \mathbb{R}^n$ denotes the new LP optimum when the constraint $\boldsymbol{\alpha}^T \boldsymbol{x} \leq \beta$ is added, we pin down a precise characterization of $\boldsymbol{x}^*_{\mathsf{LP}}(\boldsymbol{\alpha}^T \boldsymbol{x} \leq \beta)$ as a function of $\boldsymbol{\alpha}$ and $\beta$. We show that $\boldsymbol{x}^*_{\mathsf{LP}}(\boldsymbol{\alpha}^T \boldsymbol{x} \leq \beta)$ has a piece-wise closed form: there are surfaces partitioning $\mathbb{R}^{n+1}$ such that within each connected component induced by these surfaces, $\boldsymbol{x}^*_{\mathsf{LP}}(\boldsymbol{\alpha}^T \boldsymbol{x} \leq \beta)$ has a closed form. While the geometric intuition used to establish this piece-wise structure relies on the basic property that optimal solutions to LPs are achieved at vertices, the surfaces defining the regions are perhaps surprisingly nonlinear: they are defined by multivariate degree-2 polynomials in $\boldsymbol{\alpha}, \beta$. In Appendix B.1 we illustrate these surfaces for an example LP.

The proof requires us to: (1) track the set of edges of the LP polytope intersected by the new constraint, and once those edges are fixed, (2) track which edge yields the vertex with the highest objective.

Let $M = [m]$ denote the set of $m$ constraints. For $E \subseteq M$, let $A_E \in \mathbb{R}^{|E| \times n}$ and $\boldsymbol{b}_E \in \mathbb{R}^{|E|}$ denote the restrictions of $A$ and $\boldsymbol{b}$ to $E$. For $\boldsymbol{\alpha} \in \mathbb{R}^n$, $\beta \in \mathbb{R}$, and $E \subseteq M$ with $|E| = n - 1$, let $A_{E,\boldsymbol{\alpha}} \in \mathbb{R}^{n \times n}$ denote the matrix obtained by adding row vector $\boldsymbol{\alpha}$ to $A_E$ and let $A_{E,\boldsymbol{\alpha},\beta}^i \in \mathbb{R}^{n \times n}$ be the matrix $A_{E,\boldsymbol{\alpha}}$ with the $i$th column replaced by $(\boldsymbol{b}_E, \beta)^T$.

**Theorem 3.1.** *Let $(\boldsymbol{c}, A, \boldsymbol{b})$ be an LP with optimal solution $\boldsymbol{x}_{\mathsf{LP}}^*$. There are at most $m^n$ hyperplanes and $m^{2n}$ degree-2 polynomial hypersurfaces partitioning $\mathbb{R}^{n+1}$ into connected components such that for each component $C$, either: (1) $\boldsymbol{x}_{\mathsf{LP}}^*(\boldsymbol{\alpha}^T \boldsymbol{x} \leq \beta) = \boldsymbol{x}_{\mathsf{LP}}^*$, or (2) there is a set of constraints $E \subseteq M$ with $|E| = n - 1$ such that $\boldsymbol{x}_{\mathsf{LP}}^*(\boldsymbol{\alpha}^T \boldsymbol{x} \leq \beta)[i] = \det(A_{E,\boldsymbol{\alpha},\beta}^i) / \det(A_{E,\boldsymbol{\alpha}})$ for all $(\boldsymbol{\alpha}, \beta) \in C$.*

*Proof.* First, if $\boldsymbol{\alpha}^T \boldsymbol{x} \leq \beta$ does not separate $\boldsymbol{x}_{\mathsf{LP}}^*$, then $\boldsymbol{x}_{\mathsf{LP}}^*(\boldsymbol{\alpha}^T \boldsymbol{x} \leq \beta) = \boldsymbol{x}_{\mathsf{LP}}^*$. The set of all such cuts is the halfspace given by $\{(\boldsymbol{\alpha}, \beta) \in \mathbb{R}^{n+1} : \boldsymbol{\alpha}^T \boldsymbol{x}_{\mathsf{LP}}^* \leq \beta\}$. All other cuts separate $\boldsymbol{x}_{\mathsf{LP}}^*$ and thus pass through $\mathcal{P} = \{\boldsymbol{x} \in \mathbb{R}^n : A\boldsymbol{x} \leq \boldsymbol{b}, \boldsymbol{x} \geq \boldsymbol{0}\}$, and the new LP optimum is achieved at a vertex created by the cut. We consider the new vertices formed by the cut, which lie on edges of $\mathcal{P}$. Each edge $e$ of $\mathcal{P}$ can be identified with a subset $E \subset M$ of size $n - 1$ such that the edge is the set of all points $\boldsymbol{x}$ such that $\boldsymbol{a}_i^T \boldsymbol{x} = b_i$ for all $i \in E$ and $\boldsymbol{a}_i^T \boldsymbol{x} \leq b_i$ for all $i \in M \setminus E$ where $\boldsymbol{a}_i$ is the $i$th row of $A$. If we drop the inequality constraints defining the edge, the equality constraints define a line in $\mathbb{R}^n$. The intersection of the cut $\boldsymbol{\alpha}^T \boldsymbol{x} \leq \beta$ and this line is the solution to the system of $n$ linear equations in $n$ variables: $A_E \boldsymbol{x} = \boldsymbol{b}_E, \boldsymbol{\alpha}^T \boldsymbol{x} = \beta$. By Cramer's rule, the unique solution $\boldsymbol{x} = (x_1, \ldots, x_n)$ to this system is given by $x_i = \det(A_{E,\boldsymbol{\alpha},\beta}^i) / \det(A_{E,\boldsymbol{\alpha}})$. To ensure that the intersection point lies on the edge of the polytope, we stipulate that it satisfies the inequality constraints in $M \setminus E$. That is,

$$\sum_{j=1}^{n} a_{ij} \cdot \frac{\det(A_{E,\boldsymbol{\alpha},\beta}^j)}{\det(A_{E,\boldsymbol{\alpha}})} \leq b_i \tag{2}$$

for every $i \in M \setminus E$ (if $\boldsymbol{\alpha}, \beta$ satisfy any of these constraints, it must be that $\det(A_{E,\boldsymbol{\alpha}}) \neq 0$, which guarantees that $A_E \boldsymbol{x} = \boldsymbol{b}_E, \boldsymbol{\alpha}^T \boldsymbol{x} = \beta$ has a unique solution). Multiplying through by $\det(A_{E,\boldsymbol{\alpha}})$ shows that this constraint is a halfspace in $\mathbb{R}^{n+1}$, since $\det(A_{E,\boldsymbol{\alpha}})$ and $\det(A_{E,\boldsymbol{\alpha},\beta}^i)$ are linear in $\boldsymbol{\alpha}$ and $\beta$. The collection of all the hyperplanes defining the boundaries of these halfspaces over all edges of $\mathcal{P}$ induces a partition of $\mathbb{R}^{n+1}$ into connected components such that for all $(\boldsymbol{\alpha}, \beta)$ within a given component, the (nonempty) set of edges of $\mathcal{P}$ that the hyperplane $\boldsymbol{\alpha}^T \boldsymbol{x} = \beta$ intersects is invariant.

Now, consider a single connected component, denoted by $C$ for brevity. Let $e_1, \ldots, e_k$ denote the edges intersected by cuts in $C$, and let $E_1, \ldots, E_k \subset M$ denote the sets of constraints that are binding at each of these edges, respectively. For each pair $e_p, e_q$, consider the surface

$$\sum_{i=1}^{n} c_i \cdot \frac{\det(A_{E_p,\boldsymbol{\alpha},\beta}^i)}{\det(A_{E_p,\boldsymbol{\alpha}})} = \sum_{i=1}^{n} c_i \cdot \frac{\det(A_{E_q,\boldsymbol{\alpha},\beta}^i)}{\det(A_{E_q,\boldsymbol{\alpha}})}. \tag{3}$$

Clearing the (nonzero) denominators shows this is a degree-2 polynomial hypersurface in $\boldsymbol{\alpha}, \beta$ in $\mathbb{R}^{n+1}$. This hypersurface is the set of all $(\boldsymbol{\alpha}, \beta)$ for which the LP objective values achieved at the vertices on edges $e_p$ and $e_q$ are equal. The collection of these surfaces for each $p, q$ partitions $C$ into further connected components. Within each component $C'$, the edge containing the vertex that maximizes the objective is invariant. If this edge corresponds to binding constraints $E$, $\boldsymbol{x}_{\mathsf{LP}}^*(\boldsymbol{\alpha}^T \boldsymbol{x} \leq \beta)$ has the closed form $\boldsymbol{x}_{\mathsf{LP}}^*(\boldsymbol{\alpha}^T \boldsymbol{x} \leq \beta)[i] = \det(A_{E,\boldsymbol{\alpha},\beta}^i) / \det(A_{E,\boldsymbol{\alpha}})$ for all $(\boldsymbol{\alpha}, \beta) \in C'$. We now count the number of surfaces in our decomposition. $\mathcal{P}$ has at most $\binom{m}{n-1} \leq m^{n-1}$ edges, and for each edge $E$, Equation (2) defines at most $|M \setminus E| \leq m$ hyperplanes for a total of at most $m^n$ hyperplanes. Equation (3) defines a degree-2 polynomial hypersurface for every pair of edges, of which there are at most $\binom{m^n}{2} \leq m^{2n}$. $\qquad \square$

In Appendix B.2, we generalize Theorem 3.1 to understand $\boldsymbol{x}_{\mathsf{LP}}^*$ as a function of any $K$ constraints. In this case, we show that the piecewise structure is given by degree-$2K$ multivariate polynomials.

## 4 Structure and sensitivity of branch-and-cut

We now use Theorem 3.1 to answer a fundamental question about B&C: what is the structure of the B&C tree as a function of cuts at the root? Answering this question brings us one step closer toward

providing sample complexity guarantees for GMI cuts. Said another way, we derive conditions on $\boldsymbol{\alpha}_1, \boldsymbol{\alpha}_2 \in \mathbb{R}^n$, $\beta_1, \beta_2 \in \mathbb{R}$, such that B&C behaves identically on the two IPs

$$\max\{\boldsymbol{c}^T\boldsymbol{x} : A\boldsymbol{x} \leq \boldsymbol{b}, \boldsymbol{\alpha}_1^T\boldsymbol{x} \leq \beta_1, \boldsymbol{x} \in \mathbb{Z}_{\geq 0}^n\} \text{ and } \max\{\boldsymbol{c}^T\boldsymbol{x} : A\boldsymbol{x} \leq \boldsymbol{b}, \boldsymbol{\alpha}_2^T\boldsymbol{x} \leq \beta_2, \boldsymbol{x} \in \mathbb{Z}_{\geq 0}^n\}.$$

We prove that the set of all cuts can be partitioned into a finite number of regions where by employing cuts from any one region, the B&C tree remains exactly the same. We also prove that the boundaries between regions are defined by low-degree polynomials. Figure 2 is a schematic diagram of our proof, which breaks the analysis of B&C into four main steps. Each step successively refines the partition obtained in the previous step, and uses the properties established in the previous step to analyze the next stage of B&C. We focus on a single cut added to the root and extend to multiple cuts in Appendix C.2. The full proofs from this section are in Appendix C.

We use the following notation in this section. Given an IP, let $\tau = \lceil \max_{\boldsymbol{x} \in \mathcal{P}} \|\boldsymbol{x}\|_\infty \rceil$ be the maximum magnitude coordinate of any LP-feasible solution, rounded up. By Cramer's rule and Hadamard's inequality, $\tau \leq a^n n^{n/2}$ where $a = \|A\|_{\infty,\infty}$. However, $\tau$ can be much smaller. For example, if $A$ contains a row with only positive entries, then $\tau \leq \|\boldsymbol{b}\|_\infty$. Let $\mathcal{BC} := \{\boldsymbol{x}[i] \leq \ell, \boldsymbol{x}[i] \geq \ell\}_{0 \leq \ell \leq \tau, i \in [n]}$, which contains the set of all possible branching constraints. Let $A_\sigma$ and $\boldsymbol{b}_\sigma$ denote $A$ and $b$ with the constraints in $\sigma \subseteq \mathcal{BC}$ added. For $E \subseteq M \cup \sigma$, let $A_{E,\sigma} \in \mathbb{R}^{|E| \times n}$ and $\boldsymbol{b}_E \in \mathbb{R}^{|E|}$ denote the restrictions of $A_\sigma$ and $\boldsymbol{b}_\sigma$ to $E$. For $\boldsymbol{\alpha} \in \mathbb{R}^n$, $\beta \in \mathbb{R}$ and $E \subseteq M \cup \sigma$ with $|E| = n - 1$, let $A_{E,\boldsymbol{\alpha},\sigma} \in \mathbb{R}^{n \times n}$ denote the matrix obtained by adding row vector $\boldsymbol{\alpha}$ to $A_{E,\sigma}$ and let $A_{E,\boldsymbol{\alpha},\beta,\sigma}^i \in \mathbb{R}^{n \times n}$ be the matrix $A_{E,\boldsymbol{\alpha},\sigma}$ with the $i$th column replaced by $(\boldsymbol{b}_{E,\sigma}, \beta)^T$.

## 4.1 Step 1: Understanding how the cut affects the LP optimum at any node of the B&C tree

Theorem 3.1 gives a (piecewise) closed form for the LP optimum $\boldsymbol{x}_{\mathsf{LP}}^*(\boldsymbol{\alpha}^T\boldsymbol{x} \leq \beta)$ at the root of the B&C tree as a function of coefficients $(\boldsymbol{\alpha}, \beta) \in \mathbb{R}^{n+1}$ determining the cut. The first step is to extend this result to get a handle on the LP optimum at any node of any B&C tree. Suppose $\sigma \subseteq \mathcal{BC}$ is a set of branching constraints (any node of any B&C tree can be identified with some $\sigma \subseteq \mathcal{BC}$). We refine the partition of space obtained in Theorem 3.1 so that within a given region of the new partition, $\boldsymbol{x}_{\mathsf{LP}}^*(\boldsymbol{\alpha}^T\boldsymbol{x} \leq \beta, \sigma)$ has a closed form for all $\sigma$. This is illustrated by Figure 2a.

**Lemma 4.1.** *For any IP $(\boldsymbol{c}, A, \boldsymbol{b})$, there are at most $(m + 2n)^n\tau^{3n}$ hyperplanes and at most $(m + 2n)^{2n}\tau^{3n}$ degree-2 polynomial hypersurfaces partitioning $\mathbb{R}^{n+1}$ into connected components such that for each component $C$ and every $\sigma \subset \mathcal{BC}$, either: (1) $\boldsymbol{x}_{\mathsf{LP}}^*(\boldsymbol{\alpha}^T\boldsymbol{x} \leq \beta, \sigma) = \boldsymbol{x}_{\mathsf{LP}}^*(\sigma)$ and $z_{\mathsf{LP}}^*(\boldsymbol{\alpha}^T\boldsymbol{x} \leq \beta, \sigma) = z_{\mathsf{LP}}^*(\sigma)$, or (2) there is a set of constraints $E \subseteq M \cup \sigma$ with $|E| = n - 1$ such that $\boldsymbol{x}_{\mathsf{LP}}^*(\boldsymbol{\alpha}^T\boldsymbol{x} \leq \beta, \sigma)[i] = \frac{\det(A_{E,\boldsymbol{\alpha},\beta,\sigma}^i)}{\det(A_{E,\boldsymbol{\alpha},\sigma})}$ for all $(\boldsymbol{\alpha}, \beta) \in C$.*

## 4.2 Step 2: Conditions for branching decisions to be identical

We next refine the decomposition obtained in Lemma 4.1 so that the branching constraints added at each step of B&C are invariant within a region, as in Figure 2b. For concreteness, we analyze the product scoring rule (Def. 2.2) used by the leading open-source solver SCIP [14]. The high-level intuition is that we zoom in on a connected component in the partition of Lemma 4.1. Within this component, we may express $\boldsymbol{x}_{\mathsf{LP}}^*(\boldsymbol{\alpha}^T\boldsymbol{x} \leq \beta, \sigma)$ explicitly in terms of $\boldsymbol{\alpha}, \beta$, for all $\sigma$. This allows us to unravel the branching rule and derive conditions for invariance.

**Lemma 4.2.** *For any IP $(\boldsymbol{c}, A, \boldsymbol{b})$, there are at most $3(m + 2n)^n\tau^{3n}$ hyperplanes, $3(m + 2n)^{3n}\tau^{4n}$ degree-2 polynomial hypersurfaces, and $(m + 2n)^{6n}\tau^{4n}$ degree-5 polynomial hypersurfaces partitioning $\mathbb{R}^{n+1}$ into connected components such that within each component, the branching constraints used at every step of B&C are invariant.*

*Proof sketch.* If we are at a node of B&C represented by $\sigma$, the new branching constraints after expanding that node are of the form $x_i \leq \lfloor \boldsymbol{x}_{\mathsf{LP}}^*(\boldsymbol{\alpha}^T\boldsymbol{x} \leq \beta, \sigma)[i] \rfloor$ and $x_i \geq \lceil \boldsymbol{x}_{\mathsf{LP}}^*(\boldsymbol{\alpha}^T\boldsymbol{x} \leq \beta, \sigma)[i] \rceil$. Lemma 4.1 gives closed forms for the right-hand-sides of these two constraints, allowing us to control the rounding aspect of the constraints. The rest of the proof is a careful analysis of the product scoring rule which allows us to derive conditions ensuring that the branching variable is invariant. $\square$

### 4.3 Step 3: When do nodes have an integral LP optimum?

We now move to the most critical phase of branch-and-cut: deciding when to fathom a node. The first reason a node might be fathomed is if the LP relaxation of the IP at that node has an integral solution. We derive conditions that ensure that nearby cuts have the same effect on the integrality of the IP at any node in the search tree. Recall $\mathcal{P}_\mathsf{I} = \mathcal{P} \cap \mathbb{Z}^n$ is the set of integer points in $\mathcal{P}$.

**Lemma 4.3.** *For any IP $(\boldsymbol{c}, A, \boldsymbol{b})$, there are at most $3(m + 2n)^n \tau^{4n}$ hyperplanes, $3(m + 2n)^{3n} \tau^{4n}$ degree-2 polynomial hypersurfaces, and $(m + 2n)^{6n} \tau^{4n}$ degree-5 polynomial hypersurfaces partitioning $\mathbb{R}^{n+1}$ into connected components such that for each component $C$ and each $\sigma \subseteq \mathcal{BC}$, $\mathbf{1}\left[\boldsymbol{x}_\mathsf{LP}^*\left(\boldsymbol{\alpha}^T \boldsymbol{x} \le \beta, \sigma\right) \in \mathbb{Z}^n\right]$ is invariant for all $(\boldsymbol{\alpha}, \beta) \in C$.*

*Proof sketch.* For all $\sigma, \boldsymbol{x}_\mathsf{I} \in \mathcal{P}_\mathsf{I}$, and $i \in [n]$, consider the surface $\boldsymbol{x}_\mathsf{LP}^*(\boldsymbol{\alpha}^T \boldsymbol{x} \le \beta, \sigma)[i] = \boldsymbol{x}_\mathsf{I}[i]$. By Lemma 4.1, this surface is a hyperplane. If $\boldsymbol{x}_\mathsf{LP}^*(\boldsymbol{\alpha}^T \boldsymbol{x} \le \beta, \sigma) \in \mathbb{Z}^n$ for some $(\boldsymbol{\alpha}, \beta)$ in a connected component induced by these hyperplanes, $\boldsymbol{x}_\mathsf{LP}^*(\boldsymbol{\alpha}^T \boldsymbol{x} \le \beta, \sigma) = \boldsymbol{x}_\mathsf{I}$ for some $\boldsymbol{x}_\mathsf{I} \in \mathcal{P}_\mathsf{I}(\sigma) \subseteq \mathcal{P}_\mathsf{I}$, which means that $\boldsymbol{x}_\mathsf{LP}^*(\boldsymbol{\alpha}^T \boldsymbol{x} \le \beta, \sigma) = \boldsymbol{x}_\mathsf{I} \in \mathbb{Z}^n$ *for all* $(\boldsymbol{\alpha}, \beta)$ in that component. $\square$

Lemma 4.3 is illustrated by Figure 2c. Next, suppose for a moment that B&C fathoms a node if and only if either the LP is infeasible or the LP optimal solution is integral—that is, the "bounding" of B&C is suppressed. In this case, the tree built by B&C is invariant within each component of the partition in Lemma 4.3. Equipped with this observation, we now analyze the full behavior of B&C.

### 4.4 Step 4: Pruning nodes with weak LP bounds

In this final step, we analyze the most important aspect of B&C: pruning nodes when the LP objective value is smaller than the best-known integral solution. Using the tools we have developed so far, expressing the question "is the LP value at a node smaller than the best-known integral solution?" becomes a simple matter of hyperplanes and halfspaces. This final step is illustrated by Figure 2d.

**Theorem 4.4.** *Given an IP $(\boldsymbol{c}, A, \boldsymbol{b})$, there is a set of at most $O(14^n (m + 2n)^{3n^2} \tau^{5n^2})$ polynomial hypersurfaces of degree $\le 5$ partitioning $\mathbb{R}^{n+1}$ into connected components such that the B&C tree built after adding the cut $\boldsymbol{\alpha}^T \boldsymbol{x} \le \beta$ at the root is invariant over all $(\boldsymbol{\alpha}, \beta)$ within a given component.*

*Proof sketch.* Let $Q_1, \ldots, Q_{i_1}, I_1, Q_{i_1+1}, \ldots, Q_{i_2}, I_2, Q_{i_2+1}, \ldots$ denote the nodes of the B&C tree in order of exploration, under the assumption that a node is pruned if and only if either the LP at that node is infeasible or the LP optimal solution is integral. Here, a node is identified by the list $\sigma$ of branching constraints added to the input IP. Nodes labeled by $Q$ are either infeasible or have fractional LP optimal solutions. Nodes labeled by $I$ have integral LP optimal solutions and are candidates for the incumbent integral solution at the point they are encountered. By Lemma 4.3, this ordered list of nodes is invariant over any connected component of our partition.

Given an node index $\ell$, let $I(\ell)$ denote the incumbent node with the highest objective value encountered up until the $\ell$th node searched by B&C, and let $z(I(\ell))$ denote its objective value. For each node $Q_\ell$, let $\sigma_\ell$ denote the branching constraints added to arrive at node $Q_\ell$. The hyperplane $z_\mathsf{LP}^*(\boldsymbol{\alpha}^T \boldsymbol{x} \le \beta, \sigma_\ell) = z(I(\ell))$ (which is a hyperplane due to Lemma 4.1) induces two regions. In one region, $z_\mathsf{LP}^*(\boldsymbol{\alpha}^T \boldsymbol{x} \le \beta, \sigma_\ell) \le z(I(\ell))$ and so the subtree rooted at $Q_\ell$ is pruned. In the other region, $z_\mathsf{LP}^*(\boldsymbol{\alpha}^T \boldsymbol{x} \le \beta, \sigma_\ell) > z(I(\ell))$, and $Q_\ell$ is branched on further. Therefore, within each component induced by all such hyperplanes for all $\ell$, the set of nodes that are pruned is invariant. Combined with the surfaces established in Lemma 4.3, these hyperplanes partition $\mathbb{R}^{n+1}$ into components such that as $(\boldsymbol{\alpha}, \beta)$ varies within a given component, the B&C tree is invariant. $\square$

## 5 Sample complexity bounds for Gomory mixed integer cuts

In this section, we show how the results from Section 4 can be used to provide sample complexity bounds for GMI cuts (Definition 2.1), parameterized by $\boldsymbol{u} \in \mathcal{U} \subseteq \mathbb{R}^m$. We assume there is an unknown, application-specific distribution $\mathcal{D}$ over IPs. The learner receives a *training set* $\mathcal{S} \sim \mathcal{D}^N$ of $N$ IPs sampled from this distribution. A *sample complexity guarantee* bounds the number of samples $N$ sufficient to ensure that for any parameter setting $\boldsymbol{u} \in \mathcal{U}$, the B&C tree size on average over $\mathcal{S}$ is close to the expected tree size. More formally, let $g_{\boldsymbol{u}}(\boldsymbol{c}, A, \boldsymbol{b})$ be the size of the tree

B&C builds given the input $(c, A, b)$ after applying the cut defined by $u$ at the root. Given $\epsilon > 0$ and $\delta \in (0, 1)$, a sample complexity guarantee bounds the number of samples $N$ sufficient to ensure that with probability $1 - \delta$ over the draw $\mathcal{S} \sim \mathcal{D}^N$, for every parameter setting $u \in \mathcal{U}$, $|\frac{1}{N} \sum_{(c,A,b) \in \mathcal{S}} g_u(c, A, b) - \mathbb{E}[g_u(c, A, b)]| \leq \epsilon$. To derive our sample complexity guarantee, we use the notion of *pseudo-dimension* [50]. Let $\mathcal{G} = \{g_u : u \in \mathcal{U}\}$. The *pseudo-dimension of $\mathcal{G}$*, denoted $\mathrm{Pdim}(\mathcal{G})$, is the largest integer $N$ for which there exist $N$ IPs $(c_1, A_1, b_1), \ldots, (c_N, A_N, b_N)$ and $N$ thresholds $r_1, \ldots, r_N \in \mathbb{R}$ such that for every binary vector $(\sigma_1, \ldots, \sigma_N) \in \{0, 1\}^N$, there exists $g_u \in \mathcal{G}$ such that $g_u(c_i, A_i, b_i) \geq r_i$ if and only if $\sigma_i = 1$. The number of samples sufficient to ensure an error of $\varepsilon$ and confidence of $1 - \delta$ is $N = O(\frac{\kappa^2}{\epsilon^2}(\mathrm{Pdim}(\mathcal{G}) + \log \frac{1}{\delta}))$ [50]. Equivalently, for a given number of samples $N$, the error-term $\varepsilon$ is at most $\kappa\sqrt{(\mathrm{Pdim}(\mathcal{G}) + \log(1/\delta))/N}$.

So far, $\alpha, \beta$ have been parameters that do not depend on the input instance $c, A, b$. Suppose now that they do: $\alpha, \beta$ are functions of $c, A, b$ and a parameter vector $u$ (as they are for GMI cuts). Despite the structure established in the previous section, if $\alpha, \beta$ can depend on $(c, A, b)$ in arbitrary ways, one cannot even hope for a finite sample complexity, illustrated by the following impossibility result. The full proofs of all results from this section are in Appendix D.

**Theorem 5.1.** *There exist functions $\alpha_{c,A,b} : \mathcal{U} \to \mathbb{R}^n$ and $\beta_{c,A,b} : \mathcal{U} \to \mathbb{R}$ such that* $\mathrm{Pdim}(\{g_u : u \in \mathcal{U}\}) = \infty$, *where $\mathcal{U}$ is any set with $|\mathcal{U}| = |\mathbb{R}|$.*

However, in the case of GMI cuts (Def. 2.1), we show that the cutting plane coefficients parameterized by $u$ are highly structured. Combining this structure with our analysis of B&C allows us to derive polynomial sample complexity bounds. We assume that $u \in [-U, U]^m$ for some $U > 0$.

Let $\alpha : [-U, U]^m \to \mathbb{R}^n$ denote the function taking GMI cut parameters $u$ to the corresponding vector of coefficients determining the resulting cutting plane, and let $\beta : [-U, U]^m \to \mathbb{R}$ denote the offset of the resulting cutting plane. So (after multiplying through by $1 - f_0$),

$$\alpha(u)[i] = \begin{cases} f_i(1 - f_0) & \text{if } f_i \leq f_0 \\ f_0(1 - f_i) & \text{if } f_i > f_0 \end{cases}$$

and $\beta(u) = f_0(1 - f_0)$ ($f_0$ and $f_i$ are functions of $u$, but we suppress this dependence for readability).

To understand the structure of B&C as a function of GMI cut parameters, we study the preimages of components $C \subseteq \mathbb{R}^{n+1}$ under the GMI coefficient maps $\alpha : [-U, U]^m \to \mathbb{R}^n$, $\beta : [-U, U]^m \to \mathbb{R}$. If $C \subseteq \mathbb{R}^{n+1}$ (as in Theorem 4.4) is such that B&C (as a function of $\alpha, \beta$) is invariant over $C$, then B&C (as a function of GMI parameter $u$) is invariant over $D := \{u : (\alpha(u), \beta(u)) \in C\}$. Our key structural insight for GMI cuts is that if $C$ is the intersection of degree-$d$ polynomial hypersurfaces in $\mathbb{R}^{n+1}$, then $D$ is the intersection of degree-$2d$ polynomial hypersurfaces in $[-U, U]^m$. We provide the high-level intuition for this result below—the formal statements and proofs are in Appendix D.

Consider some degree-$d$ polynomial $p$ in variables $y_1, \ldots, y_{n+1}$ that defines $C$, which can be written as $\sum_{T \sqsubseteq [n+1], |T| \leq d} \lambda_T \prod_{i \in T} y_i$ for some coefficients $\lambda_T \in \mathbb{R}$, where $T \sqsubseteq [n+1]$ means that $T$ is a multiset of $[n+1]$. Evaluating at $(\alpha(u), \beta(u))$, we get

$$\sum_{|T| \leq d} \lambda_T \prod_{i \in T \cap S \setminus \{n+1\}} f_i(1 - f_0) \prod_{i \in T \setminus S \setminus \{n+1\}} f_0(1 - f_i) \prod_{i \in T \cap \{n+1\}} f_0(1 - f_0).$$

Next, substitute $f_i = u^T a_i - \lfloor u^T a_i \rfloor$ and $f_0 = u^T b - \lfloor u^T b \rfloor$. Restricted to $u$ such that the floor terms round down to some fixed integers, the above expression is a polynomial in $u$ of degree $\leq 2d$. We run this procedure for every polynomial determining $C$, for every connected component $C$ in the partition of $\mathbb{R}^{n+1}$ established in Theorem 4.4 to derive our main structural result for GMI cuts.

**Lemma 5.2.** *Consider the family of GMI cuts parameterized by $u \in [-U, U]^m$. For any IP $(c, A, b)$, there are at most $O(nU^2 \|A\|_1 \|b\|_1)$ hyperplanes and $2^{O(n^2)}(m + 2n)^{O(n^3)}\tau^{O(n^3)}$ degree-10 polynomial hypersurfaces partitioning $[-U, U]^m$ into connected components such that the B&C tree built after adding the GMI cut defined by $u$ is invariant over all $u$ within a single component.*

Bounding $\mathrm{Pdim}(\{g_u : u \in [-U, U]^m\})$ is a direct application of the main theorem of Balcan et al. [9] along with standard results bounding the VC dimension of polynomial boundaries [3].

**Theorem 5.3.** *The pseudo-dimension of the class of tree-size functions $\{g_u : u \in [-U, U]^m\}$ on the domain of IPs with $\|A\|_1 \leq a$ and $\|b\|_1 \leq b$ is $O\left(m \log(abU) + mn^3 \log(m + n) + mn^3 \log \tau\right)$.*

We generalize the analysis of this section to multiple GMI cuts at the root of the B&C tree in Appendix D. We show that if $K$ GMI cuts are sequentially applied at the root, the resulting partition of the parameter space is induced by polynomials of degree $O(K^2)$.

# 6 Conclusions

In this paper, we investigated fundamental questions about integer programming: given an integer program, what is the structure of the branch-and-cut tree as a function of a set of additional feasible constraints? Through a detailed geometric and combinatorial analysis of how additional constraints affect the LP relaxation's optimal solution, we showed that the branch-and-cut tree is piecewise constant and precisely bounded the number of pieces. We showed that the structural insights that we developed could be used to prove sample complexity bounds for learning GMI cuts, one of the most important classes of general-purpose cutting planes in integer programming.

This paper opens up a variety of directions for future research. Our sensitivity analyses in Sections 3 and 4 are fairly general and a promising direction is to explore applications to other important topics in integer programming such as column generation and lifting. Another important direction is to further develop algorithmic approaches for choosing GMI (and other) cutting planes. Currently, solvers employ a subset of GMI cuts derived from the optimal simplex tableau due to computational efficiency—it would be interesting to see if the theory developed in this paper could expand the possibilities for efficient cutting plane generation.

## Acknowledgments and Disclosure of Funding

This material is based on work supported by the National Science Foundation under grants CCF-1733556, CCF-1910321, IIS-1901403, and SES-1919453, the ARO under award W911NF2010081, the Defense Advanced Research Projects Agency under cooperative agreement HR00112020003, an AWS Machine Learning Research Award, an Amazon Research Award, a Bloomberg Research Grant, and a Microsoft Research Faculty Fellowship.

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
