## A  Further details about plots

The version of the *facility location* problem we study involves a set of locations $J$ and a set of clients $C$. Facilities are to be constructed at some subset of the locations, and the clients in $C$ are served by these facilities. Each location $j \in J$ has a cost $f_j$ of being the site of a facility, and a cost $s_{c,j}$ of serving client $c \in C$. Finally, each location $j$ has a capacity $\kappa_j$ which is a limit on the number of clients $j$ can serve. The goal of the facility location problem is to arrive at a feasible set of locations for facilities and a feasible assignment of clients to these locations that minimizes the overall cost incurred.

The facility location problem can be formulated as the following $0, 1$ IP:

$$
\begin{aligned}
\text{minimize} \quad & \sum_{j \in J} f_j x_j + \sum_{j \in J} \sum_{c \in C} s_{c,j} y_{c,j} \\
\text{subject to} \quad & \sum_{j \in J} y_{c,j} = 1 && \forall\, c \in C \\
& \sum_{c \in C} y_{c,j} \le \kappa_j x_j && \forall\, j \in J \\
& y_{c,j} \in \{0,1\} && \forall\, c \in C, j \in J \\
& x_j \in \{0,1\} && \forall\, j \in J
\end{aligned}
$$

We consider the following two distributions over facility location IPs.

**First distribution**  Facility location IPs are generated by perturbing the costs and capacities of a base facility location IP. We generated the base IP with $40$ locations and $40$ clients by choosing the location costs and client-location costs uniformly at random from $[0, 100]$ and the capacities uniformly at random from $\{0, \dots, 39\}$. To sample from the distribution, we perturb this base IP by adding independent Gaussian noise with mean $0$ and standard deviation $10$ to the cost of each location, the cost of each client-location pair, and the capacity of each location.

**Second distribution**  Facility location IPs are generated by placing $80$ evenly-spaced locations along the line segment connecting the points $(0, 1/2)$ and $(1, 1/2)$ in the Cartesian plane. The location costs are all uniformly set to $1$. Then, $80$ clients are placed uniformly at random in the unit square $[0, 1]^2$. The cost $s_{c,j}$ of serving client $c$ from location $j$ is the distance between $j$ and $c$. Location capacities are chosen uniformly at random from $\{0, \dots, 43\}$.

In our experiments, we add five cuts at the root of the B&C tree. These five cuts come from the set of Chvátal-Gomory and Gomory mixed integer cuts derived from the optimal simplex tableau of the LP relaxation. The five cuts added are chosen to maximize a weighting of cutting-plane scores:

$$
\mu \cdot \texttt{score}_1 + (1 - \mu) \cdot \texttt{score}_2. \tag{4}
$$

$\texttt{score}_1$ is the *parallelism* of a cut, which intuitively measures the angle formed by the objective vector and the normal vector of the cutting plane—promoting cutting planes that are nearly parallel with the objective direction. $\texttt{score}_2$ is the *efficacy*, or depth, of a cut, which measures the perpendicular distance from the LP optimum to the cut—promoting cutting planes that are "deeper", as measured with respect to the LP optimum. More details about these scoring rules can be found in Balcan et al. [10] and references therein. Given an IP, for each $\mu \in [0, 1]$ (discretized at steps of $0.01$) we choose the five cuts among the set of Chvátal-Gomory and Gomory mixed integer cuts that maximize (4). Figures 1a and 1b display the average tree size over 1000 samples drawn from the respective distribution for each value of $\mu$ used to choose cuts at the root. We ran our experiments using the C API of IBM ILOG CPLEX 20.1.0, with default cut generation disabled, and a 64-core machine with 512 GB of RAM.

## B  Omitted results and proofs from Section 3

### B.1  Example in two dimensions

Consider the LP

$$
\max\{x + y : x \le 1, y \ge 0, y \le x\}.
$$

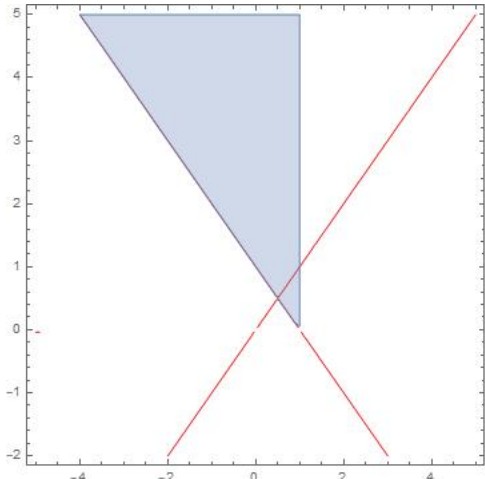

Figure 3: Decomposition of the parameter space: the blue region contains the set of $(\alpha_1, \alpha_2)$ such that the constraint intersects the feasible region at $x = 1$ and $x = y$. The red lines consist of all $(\alpha_1, \alpha_2)$ such that the objective value is equal at these intersection points. The red lines partition the blue region into two components: one where the new optimum is achieved at the intersection of $h$ and $x = y$, and one where the new optimum is achieved at the intersection of $h$ and $x = 1$.

The optimum is at $(x^*, y^*) = (1, 1)$. Consider adding an additional constraint $\alpha_1 x + \alpha_2 y \leq 1$. Let $h$ denote the hyperplane $\alpha_1 x + \alpha_2 y = 1$. We derive a description of the set of parameters $(\alpha_1, \alpha_2)$ such that $h$ intersects the hyperplanes $x = 1$ and $y = x$. The intersection of $h$ and $x = 1$ is given by

$$(x, y) = \left(1, \frac{1 - \alpha_1}{\alpha_2}\right),$$

which exists if and only if $\alpha_2 \neq 0$. This intersection point is in the LP feasible region if and only if $0 \leq \frac{1 - \alpha_1}{\alpha_2} \leq 1$ (which additionally ensures that $\alpha_2 \neq 0$). Similarly, $h$ intersects $y = x$ at

$$(x, y) = \left(\frac{1}{\alpha_1 + \alpha_2}, \frac{1}{\alpha_1 + \alpha_2}\right),$$

which exists if and only if $\alpha_1 + \alpha_2 \neq 0$. This intersection point is in the LP feasible region if and only if $0 \leq \frac{1}{\alpha_1 + \alpha_2} \leq 1$. Now, we put down an "indifference" curve in $(\alpha_1, \alpha_2)$-space that represents the set of $(\alpha_1, \alpha_2)$ such that the value of the objective achieved at the two aforementioned intersection points is equal. This surface is given by

$$\frac{2}{\alpha_1 + \alpha_2} = 1 + \frac{1 - \alpha_1}{\alpha_2}.$$

Since $\alpha_1 + \alpha_2 \neq 0$ and $\alpha_2 \neq 0$ (for the relevant $\alpha_1, \alpha_2$ in consideration), this is equivalent to $\alpha_1^2 - \alpha_2^2 - \alpha_1 + \alpha_2 = 0$, which is a degree-2 curve in $\alpha_1, \alpha_2$. The left-hand-side can be factored to write this as $(\alpha_1 - \alpha_2)(\alpha_1 + \alpha_2 - 1) = 0$. Therefore, this curve is given by the two lines $\alpha_1 = \alpha_2$ and $\alpha_1 + \alpha_2 = 1$. Figure 3 illustrates the resulting partition of $(\alpha_1, \alpha_2)$-space.

It turns out that when $n = 2$ the indifference curve can always be factored into a product of linear terms. Let the objective of the LP be $(c_1, c_2)$, and let $s_1 x + s_2 y = u_1$ and $t_1 x + t_2 y = v_1$ be two intersecting edges of the LP feasible region. Let $\alpha_1 x + \alpha_2 y = \beta$ be an additional constraint. The intersection points of this constraint with the two lines, if they exist, are given by

$$\left(\frac{s_2 \beta - u\alpha_2}{s_2 \alpha_1 - s_1 \alpha_2}, \frac{s_1 \beta - u\alpha_1}{s_1 \alpha_2 - s_2 \alpha_1}\right) \text{ and } \left(\frac{t_2 \beta - v\alpha_2}{t_2 \alpha_1 - t_1 \alpha_2}, \frac{t_2 \beta - v\alpha_1}{t_1 \alpha_2 - t_2 \alpha_1}\right).$$

The indifference surface is thus given by

$$c_1 \frac{s_2 \beta - u\alpha_2}{s_2 \alpha_1 - s_1 \alpha_2} + c_2 \frac{s_1 \beta - u\alpha_1}{s_1 \alpha_2 - s_2 \alpha_1} = c_1 \frac{t_2 \beta - v\alpha_2}{t_2 \alpha_1 - t_1 \alpha_2} + c_2 \frac{t_2 \beta - v\alpha_1}{t_1 \alpha_2 - t_2 \alpha_1}.$$

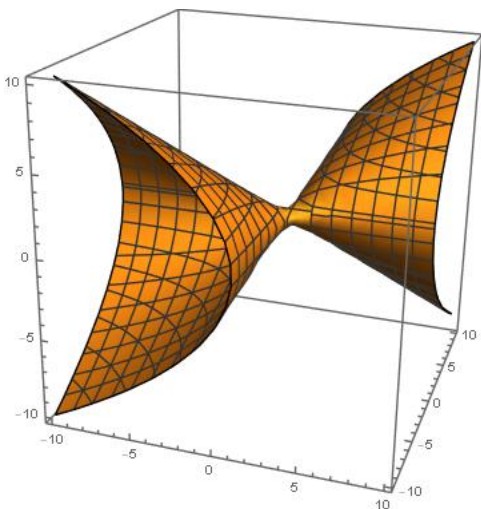

Figure 4: Indifference surface for two edges of the feasible region of an LP in three variables.

For $\alpha_1, \alpha_2$ such that $s_2\alpha_1 - s_1\alpha_2 \neq 0$ and $t_2\alpha_1 - t_1\alpha_2 \neq 0$, clearing denominators and some manipulation yields

$$(c_1\alpha_2 - c_2\alpha_1)((ut_1 - vs_1)\alpha_2 - (ut_2 - vs_2)\alpha_1 + (s_2t_2 - t_1s_2)\beta) = 0.$$

This curve consists of the two planes $c_1\alpha_2 - c_2\alpha_1 = 0$ and $(ut_1 - vs_1)\alpha_2 - (ut_2 - vs_2)\alpha_1 + (s_2t_2 - t_1s_2)\beta = 0$.

This is however not true if $n > 2$. For example, consider an LP in three variables $x, y, z$ with the constraints $x + y \leq 1, x + z \leq 1, x \leq 1, z \leq 1$. Writing out the indifference surface (assuming the objective is $\boldsymbol{c} = (1, 1, 1)^T$) for the vertex on the intersection of $\{x + y = 1, x = 1\}$ and the vertex on $\{x + z = 1, z = 1\}$ yields

$$\alpha_1\alpha_2 - \alpha_2\beta - \alpha_3^2 + \alpha_3\beta = 0.$$

Setting $\beta = 1$, we can plot the resulting surface in $\alpha_1, \alpha_2, \alpha_3$ (Figure 4).

## B.2 Linear programming sensitivity for multiple constraints

**Lemma B.1.** *Let $(\boldsymbol{c}, A, \boldsymbol{b})$ be an LP and let $M$ denote the set of its $m$ constraints. Let $\boldsymbol{x}_{\mathsf{LP}}^*$ and $z_{\mathsf{LP}}^*$ denote the optimal solution and its objective value, respectively. For $F \subseteq M$, let $A_F \in \mathbb{R}^{|F| \times n}$ and $\boldsymbol{b}_F \in \mathbb{R}^{|F|}$ denote the restrictions of $A$ and $\boldsymbol{b}$ to $F$. For $k \leq n$, $\boldsymbol{\alpha}_1, \ldots, \boldsymbol{\alpha}_k \in \mathbb{R}^n$, $\beta_1, \ldots, \beta_k \in \mathbb{R}$, and $F \subseteq M$ with $|F| = n - k$, let $A_{F, \boldsymbol{\alpha}_1, \ldots, \boldsymbol{\alpha}_k} \in \mathbb{R}^{n \times n}$ denote the matrix obtained by adding row vectors $\boldsymbol{\alpha}_1, \ldots, \boldsymbol{\alpha}_k$ to $A_F$ and let $A_{F, \boldsymbol{\alpha}_1, \beta_1, \ldots, \boldsymbol{\alpha}_k, \beta_k}^i \in \mathbb{R}^{n \times n}$ be the matrix $A_{F, \boldsymbol{\alpha}_1, \ldots, \boldsymbol{\alpha}_k} \in \mathbb{R}^{n \times n}$ with the ith column replaced by $\begin{bmatrix} \boldsymbol{b}_F & \beta_1 & \cdots & \beta_k \end{bmatrix}^T$. There is a set of at most $K$ hyperplanes, $nK^n m^n$ degree-$K$ polynomial hypersurfaces, and $nK^n m^{2n}$ degree-$2K$ polynomial hypersurfaces partitioning $\mathbb{R}^{K(n+1)}$ into connected components such that for each component $C$, one of the following holds: either (1) $\boldsymbol{x}_{\mathsf{LP}}^*(\boldsymbol{\alpha}_1^T \boldsymbol{x} \leq \beta_1, \ldots, \boldsymbol{\alpha}_K^T \boldsymbol{x} \leq \beta_K) = \boldsymbol{x}_{\mathsf{LP}}^*$, or (2) there is a subset of cuts indexed by $\ell_1, \ldots, \ell_k \in [K]$ and a set of constraints $F \subseteq M$ with $|F| = n - k$ such that*

$$\boldsymbol{x}_{\mathsf{LP}}^*(\boldsymbol{\alpha}_1^T \boldsymbol{x} \leq \beta_1, \ldots, \boldsymbol{\alpha}_K^T \boldsymbol{x} \leq \beta_K) = \left( \frac{\det(A_{F, \boldsymbol{\alpha}_{\ell_1}, \beta_{\ell_1}, \ldots, \boldsymbol{\alpha}_{\ell_k}, \beta_{\ell_k}}^1)}{\det(A_{F, \boldsymbol{\alpha}_{\ell_1}, \ldots, \boldsymbol{\alpha}_{\ell_k}})}, \ldots, \frac{\det(A_{F, \boldsymbol{\alpha}_{\ell_1}, \beta_{\ell_1}, \ldots, \boldsymbol{\alpha}_{\ell_k}, \beta_{\ell_k}}^n)}{\det(A_{F, \boldsymbol{\alpha}_{\ell_1}, \ldots, \boldsymbol{\alpha}_{\ell_k}})} \right),$$

*for all $(\boldsymbol{\alpha}_1, \beta_1, \ldots, \boldsymbol{\alpha}_K, \beta_K) \in C$.*

*Proof.* First, if none of $\boldsymbol{\alpha}_1^T \boldsymbol{x} \leq \beta_1, \ldots, \boldsymbol{\alpha}_K^T \boldsymbol{x} \leq \beta_K$ separate $\boldsymbol{x}_{\mathsf{LP}}^*$, then $\boldsymbol{x}_{\mathsf{LP}}^*(\boldsymbol{\alpha}_1^T \boldsymbol{x} \leq \beta_1, \ldots, \boldsymbol{\alpha}_K^T \boldsymbol{x} \leq \beta_K) = \boldsymbol{x}_{\mathsf{LP}}^*$ and $z_{\mathsf{LP}}^*(\boldsymbol{\alpha}_1^T \boldsymbol{x} \leq \beta_1, \ldots, \boldsymbol{\alpha}_K^T \boldsymbol{x} \leq \beta_K) = z_{\mathsf{LP}}^*$. The set of all such cuts is given by the intersection of halfspaces in $\mathbb{R}^{K(n+1)}$ given by

$$\bigcap_{j=1}^{K} \left\{ (\boldsymbol{\alpha}_1, \beta_1, \ldots, \boldsymbol{\alpha}_k, \beta_k) \in \mathbb{R}^{K(n+1)} : \boldsymbol{\alpha}_j^T \boldsymbol{x}_{\mathsf{LP}}^* \leq \beta_j \right\}. \tag{5}$$

All other vectors of $K$ cuts contain at least one cut that separates $\boldsymbol{x}^*_{\mathsf{LP}}$, and those cuts therefore pass through $\mathcal{P} = \{\boldsymbol{x} \in \mathbb{R}^n : A\boldsymbol{x} \leq \boldsymbol{b}, \boldsymbol{x} \geq \boldsymbol{0}\}$. The new LP optimum is thus achieved at a vertex created by the cuts that separate $\boldsymbol{x}^*_{\mathsf{LP}}$. As in the proof of Theorem 3.1, we consider all possible new vertices formed by our set of $K$ cuts. In the case of a single cut, these new vertices necessarily were on edges of $\mathcal{P}$, but now they may lie on higher dimensional faces.

Consider a subset of $k \leq n$ cuts that separate $\boldsymbol{x}^*_{\mathsf{LP}}$. Without loss of generality, denote these cuts by $\boldsymbol{\alpha}_1^T \boldsymbol{x} \leq \beta_1, \ldots, \boldsymbol{\alpha}_k^T \boldsymbol{x} \leq \beta_k$. We now establish conditions for these $k$ cuts to "jointly" form a new vertex of $\mathcal{P}$. Any vertex created by these cuts must lie on a face $f$ of $\mathcal{P}$ with $\dim(f) = k$ (in the case that $k = n$, the relevant face $f$ with $\dim(f) = n$ is $\mathcal{P}$ itself). Letting $M$ denote the set of $m$ constraints that define $\mathcal{P}$, each dimension-$k$ face $f$ of $\mathcal{P}$ can be identified with a (potentially empty) subset $F \subset M$ of size $n - k$ such that $f$ is precisely the set of all points $\boldsymbol{x}$ such that

$$\begin{aligned} \boldsymbol{a}_i^T \boldsymbol{x} = b_i \qquad &\forall\, i \in F \\ \boldsymbol{a}_i^T \boldsymbol{x} \leq b_i \qquad &\forall\, i \in M \setminus F, \end{aligned}$$

where $\boldsymbol{a}_i$ is the $i$th row of $A$. Let $A_F \in \mathbb{R}^{n-k \times n}$ denote the restriction of $A$ to only the rows in $F$, and let $\boldsymbol{b}_F \in \mathbb{R}^{n-k}$ denote the entries of $\boldsymbol{b}$ corresponding to the constraints in $F$. Consider removing the inequality constraints defining the face. The intersection of the cuts $\boldsymbol{\alpha}_1^T \boldsymbol{x} \leq \beta_1, \ldots, \boldsymbol{\alpha}_k^T \boldsymbol{x} \leq \beta_k$ and this unbounded surface (if it exists) is precisely the solution to the system of $n$ linear equations

$$\begin{aligned} A_F \boldsymbol{x} &= \boldsymbol{b}_F \\ \boldsymbol{\alpha}_1^T \boldsymbol{x} &= \beta_1 \\ &\vdots \\ \boldsymbol{\alpha}_k^T \boldsymbol{x} &= \beta_k. \end{aligned}$$

Let $A_{F,\boldsymbol{\alpha}_1,\ldots,\boldsymbol{\alpha}_k} \in \mathbb{R}^{n \times n}$ denote the matrix obtained by adding row vectors $\boldsymbol{\alpha}_1, \ldots, \boldsymbol{\alpha}_k$ to $A_F$, and let $A^i_{F,\boldsymbol{\alpha}_1,\beta_1,\ldots,\boldsymbol{\alpha}_k,\beta_k} \in \mathbb{R}^{n \times n}$ denote the matrix $A_{F,\boldsymbol{\alpha}_1,\ldots,\boldsymbol{\alpha}_k}$ where the $i$th column is replaced by

$$\begin{bmatrix} \boldsymbol{b}_F \\ \beta_1 \\ \vdots \\ \beta_k \end{bmatrix} \in \mathbb{R}^n.$$

By Cramer's rule, the solution to this system is given by

$$\boldsymbol{x} = \left( \frac{\det(A^1_{F,\boldsymbol{\alpha}_1,\beta_1,\ldots,\boldsymbol{\alpha}_k,\beta_k})}{\det(A_{F,\boldsymbol{\alpha}_1,\ldots,\boldsymbol{\alpha}_k})}, \ldots, \frac{\det(A^n_{F,\boldsymbol{\alpha}_1,\beta_1,\ldots,\boldsymbol{\alpha}_k,\beta_k})}{\det(A_{F,\boldsymbol{\alpha}_1,\ldots,\boldsymbol{\alpha}_k})} \right),$$

and the value of the objective at this point is

$$\boldsymbol{c}^T \boldsymbol{x} = \sum_{i=1}^n c_i \cdot \frac{\det(A^i_{F,\boldsymbol{\alpha}_1,\beta_1,\ldots,\boldsymbol{\alpha}_k,\beta_k})}{\det(A_{F,\boldsymbol{\alpha}_1,\ldots,\boldsymbol{\alpha}_k})}.$$

Now, to ensure that the unique intersection point $\boldsymbol{x}$ (1) exists and (2) actually lies on $f$ (or simply lies in $\mathcal{P}$, in the case that $F = \emptyset$), we stipulate that it satisfies the inequality constraints in $M \setminus F$. That is,

$$\sum_{j=1}^n a_{ij} \frac{\det(A^1_{F,\boldsymbol{\alpha}_1,\beta_1,\ldots,\boldsymbol{\alpha}_k,\beta_k})}{\det(A_{F,\boldsymbol{\alpha}_1,\ldots,\boldsymbol{\alpha}_k})} \leq b_i \tag{6}$$

for every $i \in M \setminus F$. If $\boldsymbol{\alpha}_1, \beta_1 \ldots, \boldsymbol{\alpha}_k, \beta_k$ satisfies any of these constraints, it must be that $\det(A_{F,\boldsymbol{\alpha}_1,\ldots,\boldsymbol{\alpha}_k}) \neq 0$, which guarantees that $A_F \boldsymbol{x} = \boldsymbol{b}_F, \boldsymbol{\alpha}_1^T \boldsymbol{x} = \beta_1, \ldots, \boldsymbol{\alpha}_k^T \boldsymbol{x} = \beta_k$ indeed has a unique solution. Now, $\det(A_{F,\boldsymbol{\alpha}_1,\ldots,\boldsymbol{\alpha}_k})$ is a polynomial in $\boldsymbol{\alpha}_1, \ldots, \boldsymbol{\alpha}_k$ of degree $\leq k$, since it is multilinear in each coefficient of each $\boldsymbol{\alpha}_\ell, \ell = 1, \ldots, k$. Similarly, $\det(A^1_{F,\boldsymbol{\alpha}_1,\beta_1,\ldots,\boldsymbol{\alpha}_k,\beta_k})$ is a polynomial in $\boldsymbol{\alpha}_1, \beta_1, \ldots, \boldsymbol{\alpha}_k, \beta_k$ of degree $\leq k$, again because it is multilinear in each cut parameter. Hence, the boundary each constraint of the form given by Equation 6 is a polynomial of degree at most $k$.

The collection of these polynomials for every $k$, every subset of $\{\boldsymbol{\alpha}_1^T \boldsymbol{x} \leq \beta_1, \ldots, \boldsymbol{\alpha}_K^T \boldsymbol{x} \leq \beta_K\}$ of size $k$, and every face of $\mathcal{P}$ of dimension $k$, along with the hyperplanes determining separation constraints (Equation 5), partition $\mathbb{R}^{K(n+1)}$ into connected components such that for all $(\boldsymbol{\alpha}_1, \beta_1, \ldots, \boldsymbol{\alpha}_K, \beta_K)$ within a given connected component, there is a fixed subset of $K$ and a fixed set of faces of $\mathcal{P}$ such that the cuts with indices in that subset intersect every face in the set at a common vertex.

Now, consider a single connected component, denoted by $C$. Let $f_1, \ldots, f_\ell$ denote the faces intersected by vectors of cuts in $C$, and let (without loss of generality) $1, \ldots, k$ denote the subset of cuts that intersect these faces. Let $F_1, \ldots, F_\ell \subset M$ denote the sets of constraints that are binding at each of these faces, respectively. For each pair $f_p, f_q$, consider the surface

$$\sum_{i=1}^n c_i \cdot \frac{\det(A_{F_p, \boldsymbol{\alpha}_1, \beta_1, \ldots, \boldsymbol{\alpha}_k, \beta_k}^i)}{\det(A_{F_p, \boldsymbol{\alpha}_1, \ldots, \boldsymbol{\alpha}_k})} = \sum_{i=1}^n c_i \cdot \frac{\det(A_{F_q, \boldsymbol{\alpha}_1, \beta_1, \ldots, \boldsymbol{\alpha}_k, \beta_k}^i)}{\det(A_{F_q, \boldsymbol{\alpha}_1, \ldots, \boldsymbol{\alpha}_k})},$$

which can be equivalently written as

$$\sum_{i=1}^n c_i \cdot \det(A_{F_p, \boldsymbol{\alpha}_1, \beta_1, \ldots, \boldsymbol{\alpha}_k, \beta_k}^i) \det(A_{F_q, \boldsymbol{\alpha}_1, \ldots, \boldsymbol{\alpha}_k}) = \sum_{i=1}^n c_i \cdot \det(A_{F_q, \boldsymbol{\alpha}_1, \beta_1, \ldots, \boldsymbol{\alpha}_k, \beta_k}^i) \det(A_{F_p, \boldsymbol{\alpha}_1, \ldots, \boldsymbol{\alpha}_k}).$$

$$(7)$$

This is a degree-$2k$ polynomial hypersurface in $(\boldsymbol{\alpha}_1, \beta_1, \ldots, \boldsymbol{\alpha}_K, \beta_K) \in \mathbb{R}^{K(n+1)}$. This hypersurface is precisely the set of all cut vectors for which the LP objective achieved at the vertex on face $f_p$ is equal to the LP objective value achieved at the vertex on face $f_q$. The collection of these surfaces for each $p, q$ partitions $C$ into further connected components. Within each of these connected components, the face containing the vertex that maximizes the objective is invariant, and the subset of cuts passing through that vertex is invariant. If $F \subseteq M$ is the set of binding constraints representing this face, and $\ell_1, \ldots, \ell_k \in [K]$ represent the subset of cuts intersecting this face, $\boldsymbol{x}_{\mathsf{LP}}^*(\boldsymbol{\alpha}_1^T \boldsymbol{x} \leq \beta_1, \ldots, \boldsymbol{\alpha}_K^T \boldsymbol{x} \leq \beta_K)$ and $z_{\mathsf{LP}}^*(\boldsymbol{\alpha}_1^T \boldsymbol{x} \leq \beta_1, \ldots, \boldsymbol{\alpha}_K^T \boldsymbol{x} \leq \beta_K)$ have the closed forms:

$$\boldsymbol{x}_{\mathsf{LP}}^*(\boldsymbol{\alpha}_1^T \boldsymbol{x} \leq \beta_1, \ldots, \boldsymbol{\alpha}_K^T \boldsymbol{x} \leq \beta_K) = \left( \frac{\det(A_{F, \boldsymbol{\alpha}_{\ell_1}, \beta_{\ell_1}, \ldots, \boldsymbol{\alpha}_{\ell_k}, \beta_{\ell_k}}^1)}{\det(A_{F, \boldsymbol{\alpha}_{\ell_1}, \ldots, \boldsymbol{\alpha}_{\ell_k}})}, \ldots, \frac{\det(A_{F, \boldsymbol{\alpha}_{\ell_1}, \beta_{\ell_1}, \ldots, \boldsymbol{\alpha}_{\ell_k}, \beta_{\ell_k}}^n)}{\det(A_{F, \boldsymbol{\alpha}_{\ell_1}, \ldots, \boldsymbol{\alpha}_{\ell_k}})} \right),$$

and

$$z_{\mathsf{LP}}^*(\boldsymbol{\alpha}_1^T \boldsymbol{x} \leq \beta_1, \ldots, \boldsymbol{\alpha}_K^T \boldsymbol{x} \leq \beta_K) = \sum_{i=1}^n c_i \cdot \frac{\det(A_{F, \boldsymbol{\alpha}_{\ell_1}, \beta_{\ell_1}, \ldots, \boldsymbol{\alpha}_{\ell_k}, \beta_{\ell_k}}^i)}{\det(A_{F, \boldsymbol{\alpha}_{\ell_1}, \ldots, \boldsymbol{\alpha}_{\ell_k}})}.$$

for all $(\boldsymbol{\alpha}_1, \beta_1, \ldots, \boldsymbol{\alpha}_K, \beta_K)$ within this component. We now count the number of surfaces used to obtain our decomposition. First, we added $K$ hyperplanes encoding separation constraints for each of the $K$ cuts (Equation 5). Then, for every subset $S \subseteq K$ of size $\leq n$, and for every face $F$ of $\mathcal{P}$ with $\dim(F) = |S|$, we first considered at most $|M \setminus F| \leq m$ degree-$\leq K$ polynomial hypersurfaces representing decision boundaries for when cuts in $S$ intersected that face (Equation 6). The number of $k$-dimensional faces of $\mathcal{P}$ is at most $\binom{m}{n-k} \leq m^{n-k} \leq m^{n-1}$, so the total number of these hypersurfaces is at most $\left( \binom{K}{0} + \cdots + \binom{K}{n} \right) m^n \leq nK^n m^n$. Finally, we considered a degree-$2K$ polynomial hypersurface for every subset of cuts and every pair of faces with degree equal to the size of the subset, of which there are at most $nK^n \binom{m^n}{2} \leq nK^n m^{2n}$. $\qquad\square$

## C   Omitted results and proofs from Section 4

We first require the following lemma which bounds the number of relevant subsets of $\mathcal{BC} := \{\boldsymbol{x}[i] \leq \ell, \boldsymbol{x}[i] \geq \ell\}_{0 \leq \ell \leq \tau, i \in [n]}$ that define a possible node expanded by B&C. $\mathcal{BC}$ is a set of size $2n(\tau + 1)$ so naïvely there are at most $2^{2n(\tau+1)}$ subsets of branching constraints. The following observation allows us to greatly reduce the number of sets we consider.

**Lemma C.1.** *Fix an IP $(\boldsymbol{c}, A, \boldsymbol{b})$. Define an equivalence relation on pairs of branching-constraint sets $\sigma_1, \sigma_2 \subseteq \mathcal{BC}$, by $\sigma_1 \sim \sigma_2 \iff \boldsymbol{x}_{\mathsf{LP}}^*(\boldsymbol{\alpha}^T \boldsymbol{x} \leq \beta, \sigma_1) = \boldsymbol{x}_{\mathsf{LP}}^*(\boldsymbol{\alpha}^T \boldsymbol{x} \leq \beta, \sigma_2)$ for all possible cutting planes $\boldsymbol{\alpha}^T \boldsymbol{x} \leq \beta$. The number of equivalence classes of $\sim$ is at most $\tau^{3n}$.*

*Proof of Lemma C.1.* Consider as an example $\sigma_1 = \{\boldsymbol{x}[1] \leq 1, \boldsymbol{x}[1] \leq 5\}$ and $\sigma_2 = \{\boldsymbol{x}[1] \leq 1\}$. We have $\boldsymbol{x}_{\mathsf{LP}}^*(\boldsymbol{\alpha}^T \boldsymbol{x} \leq \beta, \sigma_1) = \boldsymbol{x}_{\mathsf{LP}}^*(\boldsymbol{\alpha}^T \boldsymbol{x} \leq \beta, \sigma_2)$ for any cut $\boldsymbol{\alpha}^T \boldsymbol{x} \leq \beta$, because the constraint

$\boldsymbol{x}[1] \leq 5$ is redundant in $\sigma_1$. More generally, any $\sigma \subseteq \mathcal{BC}$ can be reduced by preserving only the tightest $\leq$ constraint and tightest $\geq$ constraint without affecting the resulting LP optimal solutions. The number of such unique reduced sets is at most $((\tau+2)^2)^n < \tau^{3n}$ (for each variable, there are $\tau + 2$ possibilities for the tightest $\leq$ constraint: no constraint or one of $\boldsymbol{x}[i] \leq 0, \ldots, \boldsymbol{x}[i] \leq \tau$, and similarly $\tau + 2$ possibilities for the $\geq$ constraint). $\qquad\square$

*Proof of Lemma 4.1.* We carry out the same reasoning in the proof of Theorem 3.1 for each reduced $\sigma$. The number of edges of $\mathcal{P}(\sigma)$ is at most $\binom{m+|\sigma|}{n-1} \leq (m+|\sigma|)^{n-1}$. For each edge $E$, we considered at most $|(M \cup \sigma) \setminus E| \leq m + |\sigma|$ hyperplanes, for a total of at most $(m + |\sigma|)^n$ halfspaces. Then, we had a degree-2 polynomial hypersurface for every pair of edges, of which there are at most $\binom{(m+|\sigma|)^n}{2} \leq (m + |\sigma|)^{2n}$. Summing over all reduced $\sigma$ (of which there are at most $\tau^{3n}$), combined with the fact that if $\sigma$ is reduced then $|\sigma| \leq 2n$, we get a total of at most $(m + 2n)^n \tau^{3n}$ hyperplanes and at most $(m + 2n)^{2n} \tau^{3n}$ degree-2 hypersurfaces, as desired. $\qquad\square$

Let $\mathcal{V} \subseteq \mathbb{R}^{n+1}$ denote the set of all valid cuts for the input IP $(\boldsymbol{c}, A, \boldsymbol{b})$. The set $\mathcal{V}$ is a polyhedron since it can be expressed as

$$\mathcal{V} = \bigcap_{\overline{\boldsymbol{x}} \in \mathcal{P}_\mathsf{I}} \{(\boldsymbol{\alpha}, \beta) \in \mathbb{R}^{n+1} : \boldsymbol{\alpha}^T \overline{\boldsymbol{x}} \leq \beta\},$$

and $\mathcal{P}_\mathsf{I}$ is finite as $\mathcal{P}$ is bounded. For cuts outside $\mathcal{V}$, we assume the B&C tree takes some special form denoting an invalid cut. Our goal now is to decompose $\mathcal{V}$ into connected components such that $\mathbf{1}\left[\boldsymbol{x}_\mathsf{LP}^*(\boldsymbol{\alpha}^T \boldsymbol{x} \leq \beta, \sigma) \in \mathbb{Z}^n\right]$ is invariant for all $(\boldsymbol{\alpha}, \beta)$ in each component.

*Proof of Lemma 4.3.* Fix a connected component $C$ in the decomposition that includes the facets defining $\mathcal{V}$ and the surfaces obtained in Lemma 4.2. For all $\sigma \in \mathcal{BC}$, $\boldsymbol{x}_\mathsf{I} \in \mathcal{P}_\mathsf{I}$, and $i = 1, \ldots, n$, consider the surface

$$\boldsymbol{x}_\mathsf{LP}^*(\boldsymbol{\alpha}^T \boldsymbol{x} \leq \beta, \sigma)[i] = \boldsymbol{x}_\mathsf{I}[i]. \tag{8}$$

This surface is a hyperplane, since by Lemma 4.1, either $\boldsymbol{x}_\mathsf{LP}^*(\boldsymbol{\alpha}^T \boldsymbol{x} \leq \beta, \sigma)[i] = \boldsymbol{x}_\mathsf{LP}^*(\sigma)[i]$ or $\boldsymbol{x}_\mathsf{LP}^*(\boldsymbol{\alpha}^T \boldsymbol{x} \leq \beta, \sigma)[i] = \frac{\det(A_{E,\boldsymbol{\alpha},\beta,\sigma}^i)}{\det(A_{E,\boldsymbol{\alpha},\sigma})}$, where $E \subseteq M \cup \sigma$ is the subset of constraints corresponding to $\sigma$ and $C$. Clearly, within any connected component of $C$ induced by these hyperplanes, for every $\sigma$ and $\boldsymbol{x}_\mathsf{I} \in \mathcal{P}_\mathsf{I}$, $\mathbf{1}[\boldsymbol{x}_\mathsf{LP}^*(\boldsymbol{\alpha}^T \boldsymbol{x} \leq \beta, \sigma) = \boldsymbol{x}_\mathsf{I}]$ is invariant. Finally, if $\boldsymbol{x}_\mathsf{LP}^*(\boldsymbol{\alpha}^T \boldsymbol{x} \leq \beta, \sigma) \in \mathbb{Z}^n$ for some cut $\boldsymbol{\alpha}^T \boldsymbol{x} \leq \beta$ within a given connected component, $\boldsymbol{x}_\mathsf{LP}^*(\boldsymbol{\alpha}^T \boldsymbol{x} \leq \beta, \sigma) = \boldsymbol{x}_\mathsf{I}$ for some $\boldsymbol{x}_\mathsf{I} \in \mathcal{P}_\mathsf{IH}(\sigma) \subseteq \mathcal{P}_\mathsf{I}$, which means that $\boldsymbol{x}_\mathsf{LP}^*(\boldsymbol{\alpha}^T \boldsymbol{x} \leq \beta, \sigma) = \boldsymbol{x}_\mathsf{I} \in \mathbb{Z}^n$ *for all* cuts $\boldsymbol{\alpha}^T \boldsymbol{x} \leq \beta$ in that connected component.

We now count the number of hyperplanes given by Equation 8. For each $\sigma$, there are $\binom{m+|\sigma|}{n-1} \leq (m + 2n)^{n-1}$ binding edge constraints $E \subseteq M \cup \sigma$ defining the formula of Lemma 4.1, and we have $n|\mathcal{P}_\mathsf{I}|$ hyperplanes for each $E$. Since $\tau = \max_{\boldsymbol{x} \in \mathcal{P}_\mathsf{I}} \|\boldsymbol{x}\|_\infty$, $|\mathcal{P}_\mathsf{I}| \leq \tau^n$. So the total number of hyperplanes given by Equation 8 is at most $\tau^{3n}(m + 2n)^{n-1} n \tau^n \leq (m + 2n)^n \tau^{4n}$. The number of facets defining $\mathcal{V}$ is at most $|\mathcal{P}_\mathsf{IH}| \leq |\mathcal{P}_\mathsf{I}| \leq \tau^n$. Adding these to the counts obtained in Lemma 4.2 yields the final tallies in the lemma statement. $\qquad\square$

*Proof of Theorem 4.4.* Fix a connected component $C$ in the decomposition induced by the set of hyperplanes and degree-2 hypersurfaces established in Lemma 4.3. Let

$$Q_1, \ldots, Q_{i_1}, I_1, Q_{i_1+1}, \ldots, Q_{i_2}, I_2, Q_{i_2+1}, \ldots \tag{9}$$

denote the nodes of the tree branch-and-cut creates, in order of exploration, under the assumption that a node is pruned if and only if either the LP at that node is infeasible or the LP optimal solution is integral (so the "bounding" of branch-and-bound is suppressed). Here, a node is identified by the list $\sigma$ of branching constraints added to the input IP. Nodes labeled by $Q$ are either infeasible or have fractional LP optimal solutions. Nodes labeled by $I$ have integral LP optimal solutions and are candidates for the incumbent integral solution at the point they are encountered. (The nodes are functions of $\boldsymbol{\alpha}$ and $\beta$, as are the indices $i_1, i_2, \ldots$.) By Lemma 4.3 and the observation following it, this ordered list of nodes is invariant over all $(\boldsymbol{\alpha}, \beta) \in C$.

Now, given an node index $\ell$, let $I(\ell)$ denote the incumbent node with the highest objective value encountered up until the $\ell$th node searched by B&C, and let $z(I(\ell))$ denote its objective value. For each node $Q_\ell$, let $\sigma_\ell$ denote the branching constraints added to arrive at node $Q_\ell$. The hyperplane

$$z_{\mathsf{LP}}^*(\boldsymbol{\alpha}^T\boldsymbol{x} \le \beta, \sigma_\ell) = z(I(\ell)) \tag{10}$$

(which is a hyperplane due to Lemma 4.1) partitions $C$ into two subregions. In one subregion, $z_{\mathsf{LP}}^*(\boldsymbol{\alpha}^T\boldsymbol{x} \le \beta, \sigma_\ell) \le z(I(\ell))$, that is, the objective value of the LP optimal solution is no greater than the objective value of the current incumbent integer solution, and so the subtree rooted at $Q_\ell$ is pruned. In the other subregion, $z_{\mathsf{LP}}^*(\boldsymbol{\alpha}^T\boldsymbol{x} \le \beta, \sigma_\ell) > z(I(\ell))$, and $Q_\ell$ is branched on further. Therefore, within each connected component of $C$ induced by all hyperplanes given by Equation 10 for all $\ell$, the set of node within the list (9) that are pruned is invariant. Combined with the surfaces established in Lemma 4.3, these hyperplanes partition $\mathbb{R}^{n+1}$ into connected components such that as $(\boldsymbol{\alpha}, \beta)$ varies within a given component, the tree built by branch-and-cut is invariant.

Finally, we count the total number of surfaces inducing this partition. Unlike the counting stages of the previous lemmas, we will first have to count the number of connected components induced by the surfaces established in Lemma 4.3. This is because the ordered list of nodes explored by branch-and-cut (9) can be different across each component, and the hyperplanes given by Equation 10 depend on this list. From Lemma 4.3 we have $3(m+2n)^n\tau^{4n}$ hyperplanes, $3(m+2n)^{3n}\tau^{4n}$ degree-2 polynomial hypersurfaces, and $(m+2n)^{6n}\tau^{4n}$ degree-5 polynomial hypersurfaces. To determine the connected components of $\mathbb{R}^{n+1}$ induced by the zero sets of these polynomials, it suffices to consider the zero set of the product of all polynomials defining these surfaces. Denote this product polynomial by $p$. The degree of the product polynomial is the sum of the degrees of $3(m+2n)^n\tau^{4n}$ degree-1 polynomials, $3(m+2n)^{3n}\tau^{4n}$ degree-2 polynomials, and $(m+2n)^{6n}\tau^{4n}$ degree-5 polynomials, which is at most $3(m+2n)^n\tau^{4n} + 2 \cdot 3(m+2n)^{3n}\tau^{4n} + 5 \cdot (m+2n)^{6n}\tau^{4n} < 14(m+2n)^{3n}\tau^{4n}$. By Warren's theorem, the number of connected components of $\mathbb{R}^{n+1} \setminus \{(\boldsymbol{\alpha}, \beta) : p(\boldsymbol{\alpha}, \beta) = 0\}$ is $O((14(m+2n)^{3n}\tau^{4n})^{n-1})$, and by the Milnor-Thom theorem, the number of connected components of $\{(\boldsymbol{\alpha}, \beta) : p(\boldsymbol{\alpha}, \beta) = 0\}$ is $O((14(m+2n)^{3n}\tau^{4n})^{n-1})$ as well. So, the number of connected components induced by the surfaces in Lemma 4.3 is $O(14^n(m+2n)^{3n^2}\tau^{4n^2})$. For every connected component $C$ in Lemma 4.3, the closed form of $z_{\mathsf{LP}}^*(\boldsymbol{\alpha}^T\boldsymbol{x} \le \beta, \sigma_\ell)$ is already determined due to Lemma 4.1, and so the number of hyperplanes given by Equation 10 is at most the number of possible $\sigma \subseteq \mathcal{BC}$, which is at most $\tau^{3n}$. So across all connected components $C$, the total number of hyperplanes given by Equation 10 is $O(14^n(m+2n)^{3n^2}\tau^{5n^2})$. Finally, adding this to the surface-counts established in Lemma 4.3 yields the lemma statement. $\qquad\square$

### C.1 Product scoring rule for variable selection

Let $\sigma$ be the set of branching constraints added thus far. The product scoring rule branches on the variable $i \in [n]$ that maximizes:

$$\max\{z_{\mathsf{LP}}^*(\sigma) - z_{\mathsf{LP}}^*(x_i \le \lfloor x_{\mathsf{LP}}^*(\sigma)[i]\rfloor, \sigma), \gamma\} \cdot \max\{z_{\mathsf{LP}}^*(\sigma) - z_{\mathsf{LP}}^*(x_i \ge \lceil x_{\mathsf{LP}}^*(\sigma)[i]\rceil, \sigma), \gamma\},$$

where $\gamma = 10^{-6}$.

**Lemma C.2.** *There is a set of of at most $3(m+2n)^n\tau^{3n}$ hyperplanes and $(m+2n)^{2n}\tau^{3n}$ degree-2 polynomial hypersurfaces partitioning $\mathbb{R}^{n+1}$ into connected components such that for any connected component $C$ and any $\sigma$, the set of branching constraints $\{x_i \le \lfloor x_{\mathsf{LP}}^*(\boldsymbol{\alpha}^T\boldsymbol{x} \le \beta, \sigma)[i]\rfloor, x_i \ge \lceil x_{\mathsf{LP}}^*(\boldsymbol{\alpha}^T\boldsymbol{x} \le \beta, \sigma)[i]\rceil \mid i \in [n]\}$ is invariant across all $(\boldsymbol{\alpha}, \beta) \in C$.*

*Proof.* Fix a connected component $C$ in the decomposition established in Lemma 4.1. By Lemma 4.1, for each $\sigma$, either $\boldsymbol{x}_{\mathsf{LP}}^*(\boldsymbol{\alpha}^T\boldsymbol{x} \le \beta, \sigma) = \boldsymbol{x}_{\mathsf{LP}}^*(\sigma)$ or there exists $E \subseteq M \cup \sigma$ such that $\boldsymbol{x}_{\mathsf{LP}}^*(\boldsymbol{\alpha}^T\boldsymbol{x} \le \beta, \sigma)[i] = \frac{\det(A_{E,\boldsymbol{\alpha},\beta,\sigma}^i)}{\det(A_{E,\boldsymbol{\alpha},\sigma})}$ for all $(\boldsymbol{\alpha}, \beta) \in C$. Fix a variable $i \in [n]$, which corresponds to two branching constraints

$$x_i \le \lfloor x_{\mathsf{LP}}^*(\boldsymbol{\alpha}^T\boldsymbol{x} \le \beta, \sigma)[i]\rfloor \text{ and } x_i \ge \lceil x_{\mathsf{LP}}^*(\boldsymbol{\alpha}^T\boldsymbol{x} \le \beta, \sigma)[i]\rceil. \tag{11}$$

If $C$ is a component where $\boldsymbol{x}_{\mathsf{LP}}^*(\boldsymbol{\alpha}^T\boldsymbol{x} \le \beta, \sigma) = \boldsymbol{x}_{\mathsf{LP}}^*(\sigma)$, then these two branching constraints are trivially invariant over $(\boldsymbol{\alpha}, \beta) \in C$. Otherwise, in order to further decompose $C$ such that the right-hand-sides of these constraints are invariant for every $\sigma$, we add the two decision boundaries given by

$$k \le \frac{\det(A_{E,\boldsymbol{\alpha},\beta,\sigma}^i)}{\det(A_{E,\boldsymbol{\alpha},\sigma})} \le k+1$$

for every $i$, $\sigma$, and every integer $k = 0, \ldots, \tau - 1$, where $\tau = \max_{\boldsymbol{x} \in \mathcal{P} \cap \mathbb{Z}^n} \|\boldsymbol{x}\|_\infty$. This ensures that within every connected component of $C$ induced by these boundaries (hyperplanes),

$$\left\lfloor \boldsymbol{x}^*_{\mathsf{LP}}(\boldsymbol{\alpha}^T \boldsymbol{x} \leq \beta, \sigma)[i] \right\rfloor = \left\lfloor \frac{\det(A^i_{E,\boldsymbol{\alpha},\beta,\sigma})}{\det(A_{E,\boldsymbol{\alpha},\sigma})} \right\rfloor \text{ and } \left\lceil \boldsymbol{x}^*_{\mathsf{LP}}(\boldsymbol{\alpha}^T \boldsymbol{x} \leq \beta, \sigma)[i] \right\rceil = \left\lceil \frac{\det(A^i_{E,\boldsymbol{\alpha},\beta,\sigma})}{\det(A_{E,\boldsymbol{\alpha},\sigma})} \right\rceil$$

are invariant, so the branching constraints from Equation (11) are invariant. For a fixed $\sigma$, there are two hyperplanes for every $E \subseteq M \cup \sigma$ corresponding to an edge of $\mathcal{P}(\sigma)$ and $i = 1, \ldots, n$, for a total of at most $2n\binom{m+|\sigma|}{n-1} \leq 2n(m+|\sigma|)^{n-1}$ hyperplanes. Summing over all reduced $\sigma$, we get a total of $2n(m+2n)^{n-1}\tau^{3n} < 2(m+2n)^n \tau^{3n}$ hyperplanes. Adding these hyperplanes to the set of hyperplanes established in Lemma 4.1 yields the lemma statement. $\qquad\square$

*Proof of Lemma 4.2.* Fix a connected component $C$ in the decomposition established in Lemma C.2. We know that for each set of branching constraints $\sigma$:

- By Lemma 4.1, either $\boldsymbol{x}^*_{\mathsf{LP}}(\boldsymbol{\alpha}^T \boldsymbol{x} \leq \beta, \sigma) = \boldsymbol{x}^*_{\mathsf{LP}}(\sigma)$ or there exists $E \subseteq M \cup \sigma$ such that $\boldsymbol{x}^*_{\mathsf{LP}}(\boldsymbol{\alpha}^T \boldsymbol{x} \leq \beta, \sigma)[i] = \frac{\det(A^i_{E,\boldsymbol{\alpha},\beta,\sigma})}{\det(A_{E,\boldsymbol{\alpha},\sigma})}$ for all $(\boldsymbol{\alpha}, \beta) \in C$ and all $i \in [n]$, and

- The set of branching constraints $\{x_i \leq \left\lfloor \boldsymbol{x}^*_{\mathsf{LP}}(\boldsymbol{\alpha}^T \boldsymbol{x} \leq \beta, \sigma)[i] \right\rfloor, x_i \geq \left\lceil \boldsymbol{x}^*_{\mathsf{LP}}(\boldsymbol{\alpha}^T \boldsymbol{x} \leq \beta, \sigma)[i] \right\rceil \mid i \in [n]\}$ is invariant across all $(\boldsymbol{\alpha}, \beta) \in C$.

Suppose that $\sigma$ is the list of branching constraints added so far. For any variable $k \in [n]$, let

$$\sigma_k^- = (x_k \leq \left\lfloor \boldsymbol{x}^*_{\mathsf{LP}}(\boldsymbol{\alpha}^T \boldsymbol{x} \leq \beta, \sigma)[k] \right\rfloor, \sigma) \text{ and } \sigma_k^+ = (x_k \geq \left\lceil \boldsymbol{x}^*_{\mathsf{LP}}(\boldsymbol{\alpha}^T \boldsymbol{x} \leq \beta, \sigma)[k] \right\rceil, \sigma).$$

So long as $(\boldsymbol{\alpha}, \beta) \in C$, $\sigma_k^-$ and $\sigma_k^+$ are fixed. With this notation, we can write the product scoring rule as

$$\max\{z^*_{\mathsf{LP}}(\boldsymbol{\alpha}^T \boldsymbol{x} \leq \beta, \sigma) - z^*_{\mathsf{LP}}(\boldsymbol{\alpha}^T \boldsymbol{x} \leq \beta, \sigma_k^-), \gamma\} \cdot \max\{z^*_{\mathsf{LP}}(\boldsymbol{\alpha}^T \boldsymbol{x} \leq \beta, \sigma) - z^*_{\mathsf{LP}}(\boldsymbol{\alpha}^T \boldsymbol{x} \leq \beta, \sigma_k^+), \gamma\},$$

where $\gamma = 10^{-6}$.

By Lemma 4.1, we know that across all $(\boldsymbol{\alpha}, \beta) \in C$, either $z^*_{\mathsf{LP}}(\boldsymbol{\alpha}^T \boldsymbol{x} \leq \beta, \sigma_k^+) = z^*_{\mathsf{LP}}(\sigma_k^+)$ or there exists $E_k^+ \subseteq M \cup \sigma_k^+$ such that

$$z^*_{\mathsf{LP}}\left(\boldsymbol{\alpha}^T \boldsymbol{x} \leq \beta, \sigma_k^+\right) = \sum_{i=1}^{n} c_i \cdot \frac{\det\left(A^i_{E_k^+,\boldsymbol{\alpha},\beta,\sigma_k^+}\right)}{\det\left(A_{E_k^+,\boldsymbol{\alpha},\sigma_k^+}\right)},$$

and similarly for $\sigma_k^-$, defined according to some edge set $E_k^- \subseteq M \cup \sigma_k^-$. Therefore, for each $k \in [n]$, there is a single degree-2 polynomial hypersurface partitioning $C$ into connected components such that within each connected component, either

$$z^*_{\mathsf{LP}}(\boldsymbol{\alpha}^T \boldsymbol{x} \leq \beta, \sigma) - z^*_{\mathsf{LP}}(\boldsymbol{\alpha}^T \boldsymbol{x} \leq \beta, \sigma_k^-) \geq \gamma \qquad (12)$$

or vice versa, and similarly for $\sigma_k^+$. In particular, the former hypersurface will have one of four forms:

1. $z^*_{\mathsf{LP}}(\sigma) - z^*_{\mathsf{LP}}(\sigma_k^-) \geq \gamma$, which is uniformly satisfied or not satisfied across all $(\boldsymbol{\alpha}, \beta) \in C$,

2. $z^*_{\mathsf{LP}}(\sigma) - \sum_{i=1}^{n} c_i \cdot \frac{\det\left(A^i_{E_k^-,\boldsymbol{\alpha},\beta,\sigma_k^-}\right)}{\det\left(A_{E_k^-,\boldsymbol{\alpha},\sigma_k^-}\right)} \geq \gamma$, which is a hyperplane,

3. $\sum_{i=1}^{n} c_i \cdot \frac{\det\left(A^i_{E,\boldsymbol{\alpha},\beta,\sigma}\right)}{\det(A_{E,\boldsymbol{\alpha},\sigma})} - z^*_{\mathsf{LP}}(\sigma_k^-) \geq \gamma$, which is a hyperplane, or

4. $\sum_{i=1}^{n} c_i \left( \frac{\det\left(A^i_{E,\boldsymbol{\alpha},\beta,\sigma}\right)}{\det(A_{E,\boldsymbol{\alpha},\sigma})} - \frac{\det\left(A^i_{E_k^-,\boldsymbol{\alpha},\beta,\sigma_k^-}\right)}{\det\left(A_{E_k^+,\boldsymbol{\alpha},\sigma_k^-}\right)} \right) \geq \gamma$, which is a degree-2 polynomial hypersurface.

Simply said, these are all degree-2 polynomial hypersurfaces.

Within any region induced by these hypersurfaces, the comparison between any two variables $x_k$ and $x_j$ will have the form

$$\max\{z^*_{\mathsf{LP}}(\boldsymbol{\alpha}^T\boldsymbol{x} \leq \beta, \sigma) - z^*_{\mathsf{LP}}(\boldsymbol{\alpha}^T\boldsymbol{x} \leq \beta, \sigma_k^-), \gamma\} \cdot \max\{z^*_{\mathsf{LP}}(\boldsymbol{\alpha}^T\boldsymbol{x} \leq \beta, \sigma) - z^*_{\mathsf{LP}}(\boldsymbol{\alpha}^T\boldsymbol{x} \leq \beta, \sigma_k^+), \gamma\}$$
$$\geq \max\{z^*_{\mathsf{LP}}(\boldsymbol{\alpha}^T\boldsymbol{x} \leq \beta, \sigma) - z^*_{\mathsf{LP}}(\boldsymbol{\alpha}^T\boldsymbol{x} \leq \beta, \sigma_j^-), \gamma\} \cdot \max\{z^*_{\mathsf{LP}}(\boldsymbol{\alpha}^T\boldsymbol{x} \leq \beta, \sigma) - z^*_{\mathsf{LP}}(\boldsymbol{\alpha}^T\boldsymbol{x} \leq \beta, \sigma_j^+), \gamma\}$$

which at its most complex will equal

$$\sum_{i=1}^{n} c_i \left( \frac{\det\left(A^i_{E,\boldsymbol{\alpha},\beta,\sigma}\right)}{\det\left(A_{E,\boldsymbol{\alpha},\sigma}\right)} - \frac{\det\left(A^i_{E_k^-,\boldsymbol{\alpha},\beta,\sigma_k^-}\right)}{\det\left(A_{E_k^-,\boldsymbol{\alpha},\sigma_k^-}\right)} \right) \cdot \sum_{i=1}^{n} c_i \left( \frac{\det\left(A^i_{E,\boldsymbol{\alpha},\beta,\sigma}\right)}{\det\left(A_{E,\boldsymbol{\alpha},\sigma}\right)} - \frac{\det\left(A^i_{E_k^+,\boldsymbol{\alpha},\beta,\sigma_k^+}\right)}{\det\left(A_{E_k^+,\boldsymbol{\alpha},\sigma_k^+}\right)} \right)$$

$$\tag{13}$$

$$\geq \sum_{i=1}^{n} c_i \left( \frac{\det\left(A^i_{E,\boldsymbol{\alpha},\beta,\sigma}\right)}{\det\left(A_{E,\boldsymbol{\alpha},\sigma}\right)} - \frac{\det\left(A^i_{E_j^-,\boldsymbol{\alpha},\beta,\sigma_j^-}\right)}{\det\left(A_{E_j^-,\boldsymbol{\alpha},\sigma_j^-}\right)} \right) \cdot \sum_{i=1}^{n} c_i \left( \frac{\det\left(A^i_{E,\boldsymbol{\alpha},\beta,\sigma}\right)}{\det\left(A_{E,\boldsymbol{\alpha},\sigma}\right)} - \frac{\det\left(A^i_{E_j^+,\boldsymbol{\alpha},\beta,\sigma_j^+}\right)}{\det\left(A_{E_j^+,\boldsymbol{\alpha},\sigma_j^+}\right)} \right).$$

This inequality can be written as a degree-5 polynomial hypersurface. In any region induced by these hypersurfaces, the variable that branch-and-cut branches on will be fixed.

We now count the total number of hypersurfaces. First, we count the number of degree-2 polynomial hypersurfaces from Equation (12): there is a hypersurface defined by each variable $x_k$, set of branching constraints $\sigma$, cutoff $t \in [\tau]$ such that $\sigma_k^- = (x_k \leq t, \sigma)$, set $E \subseteq M \cup \sigma$ corresponding to an edge of $\mathcal{P}(\sigma)$, and set $E_k^- \subseteq M \cup \sigma_k^-$ (and similarly for $\sigma_k^+$ and $E_k^+$). For a fixed $\sigma$, this amounts to $2n\tau\binom{m+|\sigma|}{n-1}\binom{m+|\sigma|+1}{n-1} \leq 2n\tau(m+|\sigma|+1)^{2(n-1)}$ hypersurfaces. Summing over all $\tau^{3n}$ reduced $\sigma$, we have $2n\tau^{3n+1}(m+2n+1)^{2(n-1)}$ degree-2 polynomial hypersurfaces.

Next, we count the number of degree-5 polynomial hypersurfaces from Equation (13): there is a hypersurface defined by each pair of variables $x_k, x_j$, set of branching constraints $\sigma$, cutoffs $t_k, t_j \in [\tau]$ such that $\sigma_k^- = (x_k \leq t_k, \sigma)$ and $\sigma_j^- = (x_j \leq t_j, \sigma)$, and sets $E, E_k^-, E_k^+, E_j^-, E_j^+$ corresponding to edges of $\mathcal{P}(\sigma), \mathcal{P}(\sigma_k^-), \mathcal{P}(\sigma_k^+), \mathcal{P}(\sigma_j^-), \mathcal{P}(\sigma_j^+)$. For a fixed $\sigma$, this amounts to $n^2\tau^2\binom{m+|\sigma|}{n-1}\binom{m+|\sigma|+1}{n-1}^4 \leq n^2\tau^2(m+|\sigma|+1)^{5(n-1)}$ hypersurfaces. Summing over all $\tau^{3n}$ reduced $\sigma$, we have $n^2\tau^{3n+2}(m+2n+1)^{5(n-1)}$ degree-5 polynomial hypersurfaces.

Adding these hypersurfaces to those from Lemma C.2, we get the lemma statement. $\qquad\square$

## C.2 Extension to multiple cutting planes

We can similarly derive a multi-cut version of Lemma 4.1 that controls $\boldsymbol{x}^*_{\mathsf{LP}}(\boldsymbol{\alpha}_1^T\boldsymbol{x} \leq \beta_1, \ldots, \boldsymbol{\alpha}_K^T\boldsymbol{x} \leq \beta_K, \sigma)$ for any set of branching constraints. We use the following notation. Let $(\boldsymbol{c}, A, \boldsymbol{b})$ be an LP and let $M$ denote the set of its $m$ constraints. For $F \subseteq M \cup \sigma$, let $A_{F,\sigma} \in \mathbb{R}^{|F| \times n}$ and $\boldsymbol{b}_{F,\sigma} \in \mathbb{R}^{|F|}$ denote the restrictions of $A_\sigma$ and $\boldsymbol{b}_\sigma$ to $F$. For $\boldsymbol{\alpha}_1, \ldots, \boldsymbol{\alpha}_k \in \mathbb{R}^n$, $\beta_1, \ldots, \beta_k \in \mathbb{R}$, and $F \subseteq M \cup \sigma$ with $|F| = n - k$, let $A_{F,\boldsymbol{\alpha}_1,\ldots,\boldsymbol{\alpha}_k,\sigma} \in \mathbb{R}^{n \times n}$ denote the matrix obtained by adding row vectors $\boldsymbol{\alpha}_1, \ldots, \boldsymbol{\alpha}_k$ to $A_{F,\sigma}$ and let $A^i_{F,\boldsymbol{\alpha}_1,\beta_1,\ldots,\boldsymbol{\alpha}_k,\beta_k,\sigma} \in \mathbb{R}^{n \times n}$ be the matrix $A_{F,\boldsymbol{\alpha}_1,\ldots,\boldsymbol{\alpha}_k,\sigma} \in \mathbb{R}^{n \times n}$ with the $i$th column replaced by $\begin{bmatrix} \boldsymbol{b}_{F,\sigma} & \beta_1 & \cdots & \beta_k \end{bmatrix}^T$.

**Corollary C.3.** *Fix an IP $(\boldsymbol{c}, A, \boldsymbol{b})$. There is a set of at most $K$ hyperplanes, $nK^n(m+2n)^n\tau^{3n}$ degree-$K$ polynomial hypersurfaces, and $nK^n(m+2n)^{2n}\tau^{3n}$ degree-$2K$ polynomial hypersurfaces partitioning $\mathbb{R}^{K(n+1)}$ into connected components such that for each component $C$ and every $\sigma \subseteq \mathcal{BC}$, one of the following holds: either (1) $\boldsymbol{x}^*_{\mathsf{LP}}(\boldsymbol{\alpha}_1^T\boldsymbol{x} \leq \beta_1, \ldots, \boldsymbol{\alpha}_K^T\boldsymbol{x} \leq \beta_K, \sigma) = \boldsymbol{x}^*_{\mathsf{LP}}(\sigma)$, or (2) there is a subset of cuts indexed by $\ell_1, \ldots, \ell_k \in [K]$ and a set of constraints $F \subseteq M \cup \sigma$ with $|F| = n - k$ such that*

$$\boldsymbol{x}^*_{\mathsf{LP}}(\boldsymbol{\alpha}_1^T\boldsymbol{x} \leq \beta_1, \ldots, \boldsymbol{\alpha}_K^T\boldsymbol{x} \leq \beta_K, \sigma) = \left( \frac{\det(A^1_{F,\boldsymbol{\alpha}_{\ell_1},\beta_{\ell_1},\ldots,\boldsymbol{\alpha}_{\ell_k},\beta_{\ell_k},\sigma})}{\det(A_{F,\boldsymbol{\alpha}_{\ell_1},\ldots,\boldsymbol{\alpha}_{\ell_k},\sigma})}, \ldots, \frac{\det(A^n_{F,\boldsymbol{\alpha}_{\ell_1},\beta_{\ell_1},\ldots,\boldsymbol{\alpha}_{\ell_k},\beta_{\ell_k},\sigma})}{\det(A_{F,\boldsymbol{\alpha}_{\ell_1},\ldots,\boldsymbol{\alpha}_{\ell_k},\sigma})} \right),$$

*for all $(\boldsymbol{\alpha}_1, \beta_1, \ldots, \boldsymbol{\alpha}_K, \beta_K) \in C$.*

*Proof.* The exact same reasoning in the proof of Lemma B.1 applies. We still have $K$ hyperplanes. Now, for each $\sigma$, for each subset $S \subseteq K$ with $|S| \leq n$, and for every face $F$ of $\mathcal{P}(\sigma)$ with $\dim(F) = |S|$, we have at most $m$ degree-$K$ polynomial hypersurfaces. The number of $k$-dimensional faces of $\mathcal{P}(\sigma)$ is at most $\binom{m+|\sigma|}{n-k} \leq (m+2n)^{n-1}$, so the total number of these hypersurfaces is at most $nK^n(m+2n)^n\tau^{3n}$. Finally, for every $\sigma$, we considered a degree-$2K$ polynomial hypersurfaces for every subset of cuts and every pair of faces with degree equal to the size of the subset, of which there are at most $nK^n(m+2n)^{2n}\tau^{3n}$, as desired. $\qquad\square$

We now refine the decomposition obtained in Lemma 4.1 so that the branching constraints added at each step of branch-and-cut are invariant within a region. For ease of exposition, we assume that branch-and-cut uses a lexicographic variable selection policy. This means that the variable branched on at each node of the search tree is fixed and given by the lexicographic ordering $x_1, \ldots, x_n$. Generalizing the argument to work for other policies, such as the product scoring rule, can be done as in the single-cut case.

**Lemma C.4.** *Suppose branch-and-cut uses a lexicographic variable selection policy. Then, there is a set of of at most $K$ hyperplanes, $3n^2K^n(m+2n)^n\tau^{3n}$ degree-$K$ polynomial hypersurfaces, and $nK^n(m+2n)^{2n}\tau^{3n}$ degree-$2K$ polynomial hypersurfaces partitioning $\mathbb{R}^{n+1}$ into connected components such that within each connected component, the branching constraints used at every step of branch-and-cut are invariant.*

*Proof.* Fix a connected component $C$ in the decomposition established in Corollary C.3. Then, by Corollary C.3, for each $\sigma$, either $\boldsymbol{x}^*_{\mathsf{LP}}(\boldsymbol{\alpha}_1^T\boldsymbol{x} \leq \beta_1, \ldots, \boldsymbol{\alpha}_K^T\boldsymbol{x} \leq \beta_K, \sigma) = \boldsymbol{x}^*_{\mathsf{LP}}(\sigma)$ or there exists cuts (without less of generality) labeled by indices $1, \ldots, k \in [K]$ and there exists $F \subseteq M \cup \sigma$ such that

$$\boldsymbol{x}^*_{\mathsf{LP}}(\boldsymbol{\alpha}_1^T\boldsymbol{x} \leq \beta_1, \ldots, \boldsymbol{\alpha}_K^T\boldsymbol{x} \leq \beta_K, \sigma)[i] = \frac{\det(A^i_{F,\boldsymbol{\alpha}_1,\beta_1,\ldots,\boldsymbol{\alpha}_k,\beta_k,\sigma})}{\det(A_{F,\boldsymbol{\alpha}_1,\ldots,\boldsymbol{\alpha}_k,\sigma})}$$

for all $(\boldsymbol{\alpha}, \beta) \in C$ and all $i \in [n]$. Now, if we are at a stage in the branch-and-cut tree where $\sigma$ is the list of branching constraints added so far, and the $i$th variable is being branched on next, the two constraints generated are

$$x_i \leq \left\lfloor \boldsymbol{x}^*_{\mathsf{LP}}(\boldsymbol{\alpha}_1^T\boldsymbol{x} \leq \beta_1, \ldots, \boldsymbol{\alpha}_K^T\boldsymbol{x} \leq \beta_K, \sigma)[i] \right\rfloor \text{ and } x_i \geq \left\lceil \boldsymbol{x}^*_{\mathsf{LP}}(\boldsymbol{\alpha}_1^T\boldsymbol{x} \leq \beta_1, \ldots, \boldsymbol{\alpha}_K^T\boldsymbol{x} \leq \beta_K, \sigma)[i] \right\rceil,$$

respectively. If $C$ is a component where $\boldsymbol{x}^*_{\mathsf{LP}}(\boldsymbol{\alpha}_1^T\boldsymbol{x} \leq \beta_1, \ldots, \boldsymbol{\alpha}_K^T\boldsymbol{x} \leq \beta_K, \sigma) = \boldsymbol{x}^*_{\mathsf{LP}}(\sigma)$, then there is nothing more to do, since the branching constraints at that point are trivially invariant over $(\boldsymbol{\alpha}_1, \beta_1, \ldots, \boldsymbol{\alpha}_K, \beta_K) \in C$. Otherwise, in order to further decompose $C$ such that the right-hand-side of these constraints are invariant for every $\sigma$ and every $i = 1, \ldots, n$, we add the two decision boundaries given by

$$k \leq \frac{\det(A^i_{F,\boldsymbol{\alpha}_1,\beta_1,\ldots,\boldsymbol{\alpha}_k,\beta_k,\sigma})}{\det(A_{F,\boldsymbol{\alpha}_1,\ldots,\boldsymbol{\alpha}_k,\sigma})} \leq k+1$$

for every $i$, $\sigma$, and every integer $k = 0, \ldots, \tau - 1$, where $\tau = \lceil \max_{\boldsymbol{x}\in\mathcal{P}} \|\boldsymbol{x}\|_\infty \rceil$. This ensures that within every connected component of $C$ induced by these boundaries (degree-$K$ polynomial hypersurfaces),

$$\left\lfloor \boldsymbol{x}^*_{\mathsf{LP}}(\boldsymbol{\alpha}^T\boldsymbol{x} \leq \beta, \sigma)[i] \right\rfloor = \left\lfloor \frac{\det(A^i_{F,\boldsymbol{\alpha}_1,\beta_1,\ldots,\boldsymbol{\alpha}_k,\beta_k,\sigma})}{\det(A_{F,\boldsymbol{\alpha}_1,\ldots,\boldsymbol{\alpha}_k,\sigma})} \right\rfloor$$

and

$$\left\lceil \boldsymbol{x}^*_{\mathsf{LP}}(\boldsymbol{\alpha}^T\boldsymbol{x} \leq \beta, \sigma)[i] \right\rceil = \left\lceil \frac{\det(A^i_{F,\boldsymbol{\alpha}_1,\beta_1,\ldots,\boldsymbol{\alpha}_k,\beta_k,\sigma})}{\det(A_{F,\boldsymbol{\alpha}_1,\ldots,\boldsymbol{\alpha}_k,\sigma})} \right\rceil$$

are invariant, so the branching constraints added by, for example, a lexicographic branching rule, are invariant. For a fixed $\sigma$, there are two hypersurfaces for every subset $S \subseteq [K]$, every $F \subseteq M \cup \sigma$ corresponding to a $|S|$-dimensional face of $\mathcal{P}(\sigma)$, and every $i = 1, \ldots, n$, for a total of at most $2n^2K^n\binom{m+|\sigma|}{|S|} \leq 2n^2K^n(m+2n)^n$. Summing over all reduced $\sigma$, we get a total of $2n^2K^n(m+2n)^n\tau^{3n}$ hypersurfaces. Adding these hypersurfaces to the set of hypersurfaces established in Corollary C.3 yields the lemma statement. $\qquad\square$

Now, as in the single-cut case, we consider the constraints that ensure that all cuts are valid. Let $\mathcal{V} \subseteq \mathbb{R}^{K(n+1)}$ denote the set of all vectors of valid $K$ cuts. As before, $\mathcal{V}$ is a polyhedron, since we may write

$$\mathcal{V} = \bigcap_{k=1}^{K} \bigcap_{\boldsymbol{x}_{\text{IH}} \in \mathcal{P}_{\text{IH}}} \left\{ (\boldsymbol{\alpha}_1, \beta_1, \ldots, \boldsymbol{\alpha}_K, \beta_k) \in \mathbb{R}^{K(n+1)} : \boldsymbol{\alpha}_k^T \boldsymbol{x}_{\text{IH}} \leq \beta_k \right\}.$$

We now refine our decomposition further to control the integrality of the various LP solutions at each node of branch-and-cut.

**Lemma C.5.** *Given an IP $(\boldsymbol{c}, A, \boldsymbol{b})$, there is a set of at most $2K\tau^n$ hyperplanes, $4n^2 K^n (m+2n)^n \tau^{4n}$ degree-$K$ polynomial hypersurfaces, and $nK^n(m+2n)^{2n}\tau^{3n}$ degree-$2K$ polynomial hypersurfaces partitioning $\mathbb{R}^{K(n+1)}$ into connected components such that for each component $C$, and each $\sigma \subseteq \mathcal{BC}$,*

$$\mathbf{1}\left[ \boldsymbol{x}_{\text{LP}}^* \left( \boldsymbol{\alpha}_1^T \boldsymbol{x} \leq \beta_1, \ldots, \boldsymbol{\alpha}_K^T \boldsymbol{x} \leq \beta_K, \sigma \right) \in \mathbb{Z}^n \right]$$

*is invariant for all $(\boldsymbol{\alpha}_1, \beta_1, \ldots, \boldsymbol{\alpha}_K, \beta_K) \in C$.*

*Proof.* Fix a connected component $C$ in the decomposition that includes the facets defining $\mathcal{V}$ and the surfaces obtained in Lemma C.4. For all $\sigma \in \mathcal{BC}$, $\boldsymbol{x}_{\text{I}} \in \mathcal{P}_{\text{I}}$, and $i = 1, \ldots, n$, consider the surface

$$\boldsymbol{x}_{\text{LP}}^* \left( \boldsymbol{\alpha}_1^T \boldsymbol{x} \leq \beta_1, \ldots, \boldsymbol{\alpha}_K^T \boldsymbol{x} \leq \beta_K, \sigma \right)[i] = \boldsymbol{x}_{\text{I}}[i]. \tag{14}$$

This surface is a polynomial hypersurface of degree at most $K$, due to Corollary C.3. Clearly, within any connected component of $C$ induced by these hyperplanes, for every $\sigma$ and $\boldsymbol{x}_{\text{I}} \in \mathcal{P}_{\text{I}}$, $\mathbf{1}[\boldsymbol{x}_{\text{LP}}^*(\boldsymbol{\alpha}_1^T \boldsymbol{x} \leq \beta_1, \ldots, \boldsymbol{\alpha}_K^T \boldsymbol{x} \leq \beta_K, \sigma) = \boldsymbol{x}_{\text{I}}]$ is invariant. Finally, if $\boldsymbol{x}_{\text{LP}}^*(\boldsymbol{\alpha}_1^T \boldsymbol{x} \leq \beta_1, \ldots, \boldsymbol{\alpha}_K^T \boldsymbol{x} \leq \beta_K, \sigma) \in \mathbb{Z}^n$ for some $K$ cuts $\boldsymbol{\alpha}_1^T \boldsymbol{x} \leq \beta_1, \ldots, \boldsymbol{\alpha}_K^T \boldsymbol{x} \leq \beta_K$ within a given connected component, $\boldsymbol{x}_{\text{LP}}^*(\boldsymbol{\alpha}_1^T \boldsymbol{x} \leq \beta_1, \ldots, \boldsymbol{\alpha}_K^T \boldsymbol{x} \leq \beta_K, \sigma) = \boldsymbol{x}_{\text{I}}$ for some $\boldsymbol{x}_{\text{I}} \in \mathcal{P}_{\text{IH}}(\sigma) \subseteq \mathcal{P}_{\text{I}}$, which means that $\boldsymbol{x}_{\text{LP}}^*(\boldsymbol{\alpha}_1^T \boldsymbol{x} \leq \beta_1, \ldots, \boldsymbol{\alpha}_K^T \boldsymbol{x} \leq \beta_K, \sigma) = \boldsymbol{x}_{\text{I}} \in \mathbb{Z}^n$ *for all* vectors of $K$ cuts $\boldsymbol{\alpha}_1^T \boldsymbol{x} \leq \beta_1, \ldots, \boldsymbol{\alpha}_K^T \boldsymbol{x} \leq \beta_K$ in that connected component.

We now count the number of hyperplanes given by Equation 14. For each $\sigma$, there are $nK^n$ possible subsets of cut indices and at most $(m + 2n)^{n-1}$ binding face constraints $F \subseteq M \cup \sigma$ defining the formula of Corollary C.3. For each subset-face pair, there are $n|\mathcal{P}_{\text{I}}| \leq n\tau^n$ degree-$K$ polynomial hypersurfaces given by Equation 14. So the total number of such hypersurfaces over all $\sigma$ is at most $\tau^{3n} n^2 K^n (m+2n)^{n-1}\tau^n$. The number of facets defining $\mathcal{V}$ is at most $K|\mathcal{P}_{\text{I}}| \leq K\tau^n$. Adding these to the counts obtained in Lemma C.4 yields the final tallies in the lemma statement. $\qquad\square$

At this point, as in the single-cut case, if the bounding aspect of branch-and-cut is suppressed, our decomposition yields connected components over which the branch-and-cut tree built is invariant. We now prove our main structural theorem for B&C as a function of multiple cutting planes at the root.

**Theorem C.6.** *Given an IP $(\boldsymbol{c}, A, \boldsymbol{b})$, there is a set of at most $O(12^n n^{2n} K^{2n^2} (m+2n)^{2n^2} \tau^{5n^2})$ polynomial hypersurfaces of degree at most $2K$ partitioning $\mathbb{R}^{K(n+1)}$ into connected components such that the branch-and-cut tree built after adding the $K$ cuts $\boldsymbol{\alpha}_1^T \boldsymbol{x} \leq \beta_1, \ldots, \boldsymbol{\alpha}_k^T \boldsymbol{x} \leq \beta_k$ at the root is invariant over all $(\boldsymbol{\alpha}_1, \beta_1, \ldots, \boldsymbol{\alpha}_K, \beta_K)$ within a given component. In particular, $f_{\boldsymbol{c}, A, \boldsymbol{b}}(\boldsymbol{\alpha}_1, \beta_1, \ldots, \boldsymbol{\alpha}_K, \beta_K)$ is invariant over each connected component.*

*Proof.* Fix a connected component $C$ in the decomposition induced by the set of hyperplanes, degree-$K$ hypersurfaces, and degree-$2K$ hypersurfaces established in Lemma C.5. Let

$$Q_1, \ldots, Q_{i_1}, I_1, Q_{i_1+1}, \ldots, Q_{i_2}, I_2, Q_{i_2+1}, \ldots \tag{15}$$

denote the nodes of the tree branch-and-cut creates, in order of exploration, under the assumption that a node is pruned if and only if either the LP at that node is infeasible or the LP optimal solution is integral (so the "bounding" of branch-and-bound is suppressed). Here, a node is identified by the list $\sigma$ of branching constraints added to the input IP. Nodes labeled by $Q$ are either infeasible or have fractional LP optimal solutions. Nodes labeled by $I$ have integral LP optimal solutions and are candidates for the incumbent integral solution at the point they are encountered. (The nodes are functions of $\boldsymbol{\alpha}_1, \beta_1, \ldots, \boldsymbol{\alpha}_K, \beta_K$, as are the indices $i_1, i_2, \ldots$.) By Lemma C.5, this ordered list of nodes is invariant for all $(\boldsymbol{\alpha}_1, \beta_1, \ldots, \boldsymbol{\alpha}_K, \beta_k) \in C$.

Now, given an node index $\ell$, let $I(\ell)$ denote the incumbent node with the highest objective value encountered up until the $\ell$th node searched by B&C, and let $z(I(\ell))$ denote its objective value. For each node $Q_\ell$, let $\sigma_\ell$ denote the branching constraints added to arrive at node $Q_\ell$. The hyperplane

$$z_{\mathsf{LP}}^*\left(\boldsymbol{\alpha}_1^T\boldsymbol{x} \leq \beta_1, \ldots, \boldsymbol{\alpha}_K^T\boldsymbol{x} \leq \beta_K, \sigma_\ell\right) = z(I(\ell)) \tag{16}$$

(which is a hyperplane due to Corollary C.3) partitions $C$ into two subregions. In one subregion, $z_{\mathsf{LP}}^*(\boldsymbol{\alpha}_1^T\boldsymbol{x} \leq \beta_1, \ldots, \boldsymbol{\alpha}_k^T\boldsymbol{x} \leq \beta_k, \sigma_\ell) \leq z(I(\ell))$, that is, the objective value of the LP optimal solution is no greater than the objective value of the current incumbent integer solution, and so the subtree rooted at $Q_\ell$ is pruned. In the other subregion, $z_{\mathsf{LP}}^*(\boldsymbol{\alpha}_1^T\boldsymbol{x} \leq \beta_1, \ldots, \boldsymbol{\alpha}_k^T\boldsymbol{x} \leq \beta_k, \sigma_\ell) > z(I(\ell))$, and $Q_\ell$ is branched on further. Therefore, within each connected component of $C$ induced by all hyperplanes given by Equation 16 for all $\ell$, the set of node within the list (15) that are pruned is invariant. Combined with the surfaces established in Lemma C.5, these hyperplanes partition $\mathbb{R}^{K(n+1)}$ into connected components such that as $(\boldsymbol{\alpha}_1, \beta_1 \ldots, \boldsymbol{\alpha}_K, \beta_K)$ varies within a given component, the tree built by branch-and-cut is invariant.

Finally, we count the total number of surfaces inducing this partition. Unlike the counting stages of the previous lemmas, we will first have to count the number of connected components induced by the surfaces established in Lemma C.5. This is because the ordered list of nodes explored by branch-and-cut (15) can be different across each component, and the hyperplanes given by Equation 16 depend on this list. From Lemma C.5 we have $6n^2K^n(m+2n)^{2n}\tau^{4n}$ polynomial hypersurfaces of degree $\leq 2K$. The set of all $(\boldsymbol{\alpha}_1, \beta_1, \ldots \boldsymbol{\alpha}_K, \beta_k) \in \mathbb{R}^{K(n+1)}$ such that $(\boldsymbol{\alpha}_1, \beta_1, \ldots, \boldsymbol{\alpha}_K, \beta_K)$ lies on the boundary of any of these surfaces is precisely the zero set of the product of all polynomials defining these surfaces. Denote this product polynomial by $p$. The degree of the product polynomial is the sum of the degrees of $6n^2K^n(m+2n)^{2n}\tau^{4n}$ polynomials of degree $\leq 2K$, which is at most $2K \cdot 6Kn^2K^n(m+2n)^{2n}\tau^{4n} = 12n^2K^{n+2}(m+2n)^{2n}\tau^{4n}$. By Warren's theorem, the number of connected components of $\mathbb{R}^{n+1} \setminus \{(\boldsymbol{\alpha}, \beta) : p(\boldsymbol{\alpha}, \beta) = 0\}$ is $O((12n^2K^{n+2}(m+2n)^{2n}\tau^{4n})^{n-1})$, and by the Milnor-Thom theorem, the number of connected components of $\{(\boldsymbol{\alpha}, \beta) : p(\boldsymbol{\alpha}, \beta) = 0\}$ is $O((12n^2K^{n+2}(m+2n)^{2n}\tau^{4n})^{n-1})$ as well. So, the number of connected components induced by the surfaces in Lemma C.5 is $O(12^n n^{2n} K^{2n^2}(m+2n)^{2n^2}\tau^{4n^2})$. For every connected component $C$ in Lemma C.5, the closed form of $z_{\mathsf{LP}}^*(\boldsymbol{\alpha}^T\boldsymbol{x} \leq \beta, \sigma_\ell)$ is already determined due to Corollary C.3, and so the number of hyperplanes given by Equation 16 is at most the number of possible $\sigma \subseteq \mathcal{BC}$, which is at most $\tau^{3n}$. So across all connected components $C$, the total number of hyperplanes given by Equation 16 is $O(12^n n^{2n} K^{2n^2}(m+2n)^{2n^2}\tau^{5n^2})$. Finally, adding this to the surface-counts established in Lemma C.5 yields the theorem statement. $\square$

# D  Omitted results from Section 5

*Proof of Theorem 5.1.* For a set $\mathcal{X}$, $\mathcal{X}^{<\mathbb{N}}$ denotes the set of finite sequences of elements from $\mathcal{X}$. There is a bijection between the set of IPs $(\boldsymbol{c}, A, \boldsymbol{b}) \in \mathcal{I} := \mathbb{R}^n \times \mathbb{Z}^{m \times n} \times \mathbb{Z}^m$ and $\mathbb{R}$, so IPs can be uniquely represented as real numbers (and vice versa). Now, consider the set of all finite sequences of pairs of IPs and $\pm 1$ labels of the form $((\boldsymbol{c_1}, A_1, \boldsymbol{b_1}), \varepsilon_1), \ldots, ((\boldsymbol{c_N}, A_N, \boldsymbol{b_N}), \varepsilon_N)$, $\varepsilon_1, \ldots, \varepsilon_N \in \{-1, 1\}$, that is, the set $(\mathcal{I} \times \{-1, 1\})^{<\mathbb{N}}$. There is a bijection between this set and $(\mathbb{R} \times \{-1, 1\})^{<\mathbb{N}}$, and in turn there is a bijection between $(\mathbb{R} \times \{-1, 1\})^{<\mathbb{N}}$ and $\mathbb{R}$. Hence, there exists a bijection between $\mathcal{U}$ and $(\mathcal{I} \times \{-1, 1\})^{<\mathbb{N}}$. Fix such a bijection $\varphi : \mathcal{U} \to (\mathcal{I} \times \{-1, 1\})^{<\mathbb{N}}$, and let $\varphi^{-1} : (\mathcal{I} \times \{-1, 1\})^{<\mathbb{N}} \to \mathcal{U}$ denote the inverse of $\varphi$, which is well defined and also a bijection.

Let $n$ be odd. For $c \in \mathbb{R}$, let $\mathsf{IP}_c \in \mathcal{I}$ denote the IP

$$\begin{aligned} \text{maximize} \quad & c \\ \text{subject to} \quad & 2x_1 + \cdots + 2x_n = n \\ & \boldsymbol{x} \in \{0, 1\}^n. \end{aligned} \tag{17}$$

Since $n$ is odd, $\mathsf{IP}_c$ is infeasible, independent of $c$. Jeroslow [32] showed that without the use of cutting planes or heuristics, branch-and-bound builds a tree of size $2^{(n-1)/2}$ before determining infeasibility and terminating. The objective $c$ is irrelevant, but is important in generating distinct IPs with this property. Consider the cut $x_1 + \cdots + x_n \leq \lfloor n/2 \rfloor$, which is a valid cut for $\mathsf{IP}_c$ (this is in fact a Chvátal-Gomory cut [10]). In particular, since $n$ is odd, $x_1 + \cdots + x_n \leq \lfloor n/2 \rfloor \implies x_1 + \cdots + x_n \leq (n-1)/2 < n/2$, so the equality constraint of $\mathsf{IP}_c$ is violated by this cut. Thus, the feasible region of

the LP relaxation after adding this cut is empty, and branch-and-bound will terminate immediately at the root (building a tree of size 1). Denote this cut by $(\boldsymbol{\alpha}^{(-1)}, \beta^{(-1)}) = (\mathbf{1}, \lfloor n/2 \rfloor)$. On the other hand, let $(\boldsymbol{\alpha}^{(1)}, \beta^{(1)}) = (\mathbf{0}, 0)$ be the trivial cut $0 \le 0$. Adding this cut to the IP constraints does not change the feasible region, so branch-and-bound will build a tree of size $2^{(n-1)/2}$.

We now define $\boldsymbol{\alpha}_{\boldsymbol{c},A,\boldsymbol{b}}$ and $\beta_{\boldsymbol{c},A,\boldsymbol{b}}$. Let

$$(\boldsymbol{\alpha}_{\boldsymbol{c},A,\boldsymbol{b}}(\boldsymbol{u}), \beta_{\boldsymbol{c},A,\boldsymbol{b}}(\boldsymbol{u})) = \begin{cases} (\boldsymbol{\alpha}^{(1)}, \beta^{(1)}) & \text{if } ((\boldsymbol{c}, A, \boldsymbol{b}), 1) \in \varphi(\boldsymbol{u}) \text{ and } ((\boldsymbol{c}, A, \boldsymbol{b}), -1) \notin \varphi(\boldsymbol{u}) \\ (\boldsymbol{\alpha}^{(-1)}, \beta^{(-1)}) & \text{if } ((\boldsymbol{c}, A, \boldsymbol{b}), -1) \in \varphi(\boldsymbol{u}) \text{ and } ((\boldsymbol{c}, A, \boldsymbol{b}), 1) \notin \varphi(\boldsymbol{u}) \\ (\mathbf{0}, 0) & \text{otherwise} \end{cases}.$$

The choice to use $(\mathbf{0}, 0)$ in the case that either $((\boldsymbol{c}, A, \boldsymbol{b}), \varepsilon) \notin \varphi(\boldsymbol{u})$ for each $\varepsilon \in \{-1, 1\}$, or $((\boldsymbol{c}, A, \boldsymbol{b}), -1) \in \varphi(\boldsymbol{u})$ and $((\boldsymbol{c}, A, \boldsymbol{b}), 1) \in \varphi(\boldsymbol{u})$ is arbitrary and unimportant. Now, for any integer $N > 0$, constructing a set of $N$ IPs and $N$ thresholds that is shattered is almost immediate. Let $c_1, \ldots, c_N \in \mathbb{R}$ be distinct reals, and let $1 < r_1, \ldots, r_N < 2^{(n-1)/2}$. Then, the set $\{(\mathsf{IP}_{c_1}, r_1), \ldots, (\mathsf{IP}_{c_N}, r_N)\}$ can be shattered. Indeed, given a sign pattern $(\varepsilon_1, \ldots, \varepsilon_N) \in \{-1, 1\}^N$, let

$$\boldsymbol{u} = \varphi^{-1}\left((\mathsf{IP}_{c_1}, \varepsilon_1), \ldots, (\mathsf{IP}_{c_N}, \varepsilon_N)\right).$$

Then, if $\varepsilon_i = 1$, $(\boldsymbol{\alpha}_{\mathsf{IP}_{c_i}}(\boldsymbol{u}), \beta_{\mathsf{IP}_{c_i}}(\boldsymbol{u})) = (\boldsymbol{\alpha}^{(1)}, \beta^{(1)})$, so $g_{\boldsymbol{u}}(\mathsf{IP}_{c_i}) = 2^{(n-1)/2}$ and $\text{sign}(g_{\boldsymbol{u}}(\mathsf{IP}_{c_i}) - r_i) = 1$. If $\varepsilon_i = -1$, $(\boldsymbol{\alpha}_{\mathsf{IP}_{c_i}}(\boldsymbol{u}), \beta_{\mathsf{IP}_{c_i}}(\boldsymbol{u})) = (\boldsymbol{\alpha}^{(-1)}, \beta^{(-1)})$, so $g_{\boldsymbol{u}}(\mathsf{IP}_{c_i}) = 1$ and $\text{sign}(g_{\boldsymbol{u}}(\mathsf{IP}_{c_i}) - r_i) = -1$. So for any $N$ there is a set of IPs and thresholds that can be shattered, which yields the theorem statement. □

**Lemma D.1.** *Consider the family of GMI cuts parameterized by $\boldsymbol{u} \in [-U, U]^m$. There is a set of at most $O(nU^2 \|A\|_1 \|\boldsymbol{b}\|_1)$ hyperplanes partitioning $[-U, U]^m$ into connected components such that $\lfloor \boldsymbol{u}^T \boldsymbol{a}_i \rfloor$, $\lfloor \boldsymbol{u}^T \boldsymbol{b} \rfloor$, and $\mathbf{1}[f_i \le f_0]$ are invariant, for every $i$, within each component.*

*Proof of Lemma D.1.* We have $f_i = \boldsymbol{u}^T \boldsymbol{a}_i - \lfloor \boldsymbol{u}^T \boldsymbol{a}_i \rfloor$, $f_0 = \boldsymbol{u}^T \boldsymbol{b} - \lfloor \boldsymbol{u}^T \boldsymbol{b} \rfloor$, and since $\boldsymbol{u} \in [-U, U]^m$, $\lfloor \boldsymbol{u}^T \boldsymbol{a}_i \rfloor \in [-U \|\boldsymbol{a}_i\|_1, U \|\boldsymbol{a}_i\|_1]$ and $\lfloor \boldsymbol{u}^T \boldsymbol{b} \rfloor \in [-U \|\boldsymbol{b}\|_1, U \|\boldsymbol{b}\|_1]$. Now, for all $i$, $k_i \in [-U \|\boldsymbol{a}_i\|_1, U \|\boldsymbol{a}_i\|_1] \cap \mathbb{Z}$ and $k_0 \in [-U \|\boldsymbol{b}\|_1, U \|\boldsymbol{b}\|_1] \cap \mathbb{Z}$, put down the hyperplanes defining the two halfspaces

$$\lfloor \boldsymbol{u}^T \boldsymbol{a}_i \rfloor = k_i \iff k_i \le \boldsymbol{u}^T \boldsymbol{a}_i < k_i + 1 \tag{18}$$

and the hyperplanes defining the two halfspaces

$$\lfloor \boldsymbol{u}^T \boldsymbol{b} \rfloor = k_0 \iff k_0 \le \boldsymbol{u}^T \boldsymbol{b} < k_0 + 1. \tag{19}$$

In addition, consider the hyperplane

$$\boldsymbol{u}^T \boldsymbol{a}_i - k_i = \boldsymbol{u}^T \boldsymbol{b} - k_0 \tag{20}$$

for each $i$. Within any connected component of $\mathbb{R}^m$ determined by these hyperplanes, $\lfloor \boldsymbol{u}^T \boldsymbol{a}_i \rfloor$ and $\lfloor \boldsymbol{u}^T \boldsymbol{b} \rfloor$ are constant. Furthermore, $\mathbf{1}[f_i \le f_0]$ is invariant within each connected component, since if $\lfloor \boldsymbol{u}^T \boldsymbol{a}_i \rfloor = k_i$ and $\lfloor \boldsymbol{u}^T \boldsymbol{b} \rfloor = k_0$, $f_i \le f_0 \iff \boldsymbol{u}^T \boldsymbol{a}_i - k_i \le \boldsymbol{u}^T \boldsymbol{b} - k_0$, which is the hyperplane given by Equation 20. The total number of hyperplanes of type 18 is $O(nU \|A\|_1)$, the total number of hyperplanes of type 19 is $O(U \|\boldsymbol{b}\|_1)$, and the total number of hyperplanes of type 20 is $nU^2 \|A\|_1 \|\boldsymbol{b}\|_1$. Summing yields the lemma statement. □

The next lemma allows us to transfer the polynomial partition of $\mathbb{R}^{n+1}$ from Theorem 4.4 to a polynomial partition of $[-U, U]^m$, incurring only a factor 2 increase in degree.

**Lemma D.2.** *Let $p \in \mathbb{R}[y_1, \ldots, y_{n+1}]$ be a polynomial of degree $d$. Let $D \subseteq [-U, U]^m$ be a connected component from Lemma D.1. Define $q : D \to \mathbb{R}$ by $q(\boldsymbol{u}) = p(\boldsymbol{\alpha}(\boldsymbol{u}), \beta(\boldsymbol{u}))$. Then $q$ is a polynomial in $\boldsymbol{u}$ of degree $2d$.*

*Proof.* By Lemma D.1, there are integers $k_0, k_i$ for $i \in [n]$ such that $\lfloor \boldsymbol{u}^T \boldsymbol{a}_i \rfloor = k_i$ and $\lfloor \boldsymbol{u}^T \boldsymbol{b} \rfloor = k_0$ for all $\boldsymbol{u} \in D$. Also, the set $S = \{i : f_i \le f_0\}$ is fixed over all $\boldsymbol{u} \in D$.

A degree-$d$ polynomial $p$ in variables $y_1, \ldots, y_{n+1}$ can be written as $\sum_{T \sqsubseteq [n+1], |T| \le d} \lambda_T \prod_{i \in T} y_i$ for some coefficients $\lambda_T \in \mathbb{R}$, where $T \sqsubseteq [n+1]$ means that $T$ is a multiset of $[n+1]$. Evaluating at $(\boldsymbol{\alpha}(\boldsymbol{u}), \beta(\boldsymbol{u}))$, we get

$$\sum_{|T| \le d} \lambda_T \prod_{\substack{i \in T \cap S \\ i \neq n+1}} f_i(1 - f_0) \prod_{\substack{i \in T \setminus S \\ i \neq n+1}} f_0(1 - f_i) \prod_{\substack{i \in T \\ i = n+1}} f_0(1 - f_0).$$

Now, $f_i = \boldsymbol{u}^T \boldsymbol{a}_i - k_i$ and $f_0 = \boldsymbol{u}^T \boldsymbol{b} - k_0$ are linear in $\boldsymbol{u}$. The sum is over all multisets of size at most $d$, so each monomial consists of the product of at most $d$ degree-2 terms of the form $f_i(1 - f_0)$, $f_0(1 - f_i)$, or $f_0(1 - f_0)$. Thus, $\deg(q) \le 2d$, as desired. $\qquad \square$

*Proof of Lemma 5.2.* Let $C \subseteq \mathbb{R}^{n+1}$ be a connected component in the partition established in Theorem 4.4, so $C$ can be written as the intersection of at most $14^n (m + 2n)^{3n^2} \tau^{5n^2}$ polynomial constraints of degree at most 5. Let $D \subseteq [-U, U]^m$ be a connected component in the partition established in Lemma D.1. By Lemma D.2, there are at most $14^n (m + 2n)^{3n^2} \tau^{5n^2}$ polynomials of degree at most 10 partitioning $D$ into connected components such that within each component, $\mathbf{1}[(\boldsymbol{\alpha}(\boldsymbol{u}), \beta(\boldsymbol{u})) \in C]$ is invariant. If we consider the overlay of these polynomial surfaces over all components $C$, we will get a partition of $[-U, U]^m$ such that *for every* $C$, $\mathbf{1}[(\boldsymbol{\alpha}(\boldsymbol{u}), \beta(\boldsymbol{u})) \in C]$ is invariant over each connected component of $[-U, U]^m$. Once we have this we are done, since all $\boldsymbol{u}$ in the same connected component of $[-U, U]^m$ will be sent to the same connected component of $\mathbb{R}^{n+1}$ by $(\boldsymbol{\alpha}(\boldsymbol{u}), \beta(\boldsymbol{u}))$, and thus by Theorem 4.4 the behavior of branch-and-cut will be invariant.

We now tally up the total number of surfaces. The number of connected components $C$ was given by Warren's theorem and the Milnor-Thom theorem to be $O(14^{n(n+1)} (m + 2n)^{3n^2(n+1)} \tau^{5n^2(n+1)})$, so the total number of degree-10 hypersurfaces is $14^n (m + 2n)^{3n^2} \tau^{5n^2}$ times this quantity, which yields the lemma statement. $\qquad \square$

## D.1 Multiple GMI cuts at the root

In this section we extend our results to allow for multiple GMI cuts at the root of the B&C tree. These cuts can be added simultaneously, sequentially, or in rounds. If GMI cuts $\boldsymbol{u}_1$, $\boldsymbol{u}_2$ are added simultaneously, both of them have the same dimension and are defined in the usual way. If GMI cuts $\boldsymbol{u}_1$, $\boldsymbol{u}_2$ are added sequentially, $\boldsymbol{u}_2$ has one more entry than $\boldsymbol{u}_1$. This is because when cuts are added sequentially, the LP relaxation is re-solved after the addition of the first cut, and the second cut has a multiplier for all original constraints as well as for the first cut (this ensures that the second cut can be chosen in a more informed manner). If $K$ cuts are made at the root, they can be added in sequential rounds of simultaneous cuts. In the following discussion, we focus on the case where all $K$ cuts are added sequentially—the other cases can be viewed as instantiations of this. We refer the reader to the discussion in Balcan et al. [10] for more details.

To prove an analogous result for multiple GMI cuts (in sequence, that is, each successive GMI cut has one more parameter than the previous), we combine the reasoning used in the single-GMI-cut case with some technical observations in Balcan et al. [10].

**Lemma D.3.** *Consider the family of $K$ sequential GMI cuts parameterized by $\boldsymbol{u}_1 \in [-U, U]^m$, $\boldsymbol{u}_2 \in [-U, U]^{m+1}, \ldots, \boldsymbol{u}_K \in [-U, U]^{m+K-1}$. For any IP $(\boldsymbol{c}, A, \boldsymbol{b})$, there are at most*

$$O\left(nK(1 + U)^{2K} \|A\|_1 \|\boldsymbol{b}\|_1\right)$$

*degree-$K$ polynomial hypersurfaces and*

$$2^{O(n^2)} K^{O(n^3)} (m + 2n)^{O(n^3)} \tau^{O(n^3)}$$

*degree-$4K^2$ polynomial hypersurfaces partitioning $[-U, U]^m \times \cdots \times [-U, U]^{m+K-1}$ connected components such that the B&C tree built after sequentially adding the GMI cuts defined by $\boldsymbol{u}_1, \ldots, \boldsymbol{u}_K$ is invariant over all $(\boldsymbol{u}_1, \ldots, \boldsymbol{u}_K)$ within a single component.*

*Proof.* We start with the setup used by Balcan et al. [10] to prove similar results for sequential Chvátal-Gomory cuts. Let $\boldsymbol{a}_1, \ldots, \boldsymbol{a}_n \in \mathbb{R}^m$ be the columns of $A$. We define the following augmented

columns $\widetilde{a}_i^1 \in \mathbb{R}^m, \ldots, \widetilde{a}_i^K \in \mathbb{R}^{m+K-1}$ for each $i \in [n]$, and the augmented constraint vectors $\widetilde{b}^1 \in \mathbb{R}^m, \ldots, \widetilde{b}^K \in \mathbb{R}^{m+K-1}$ via the following recurrences:

$$\widetilde{a}_i^1 = a_i$$

$$\widetilde{a}_i^k = \begin{bmatrix} \widetilde{a}_i^{k-1} \\ u_{k-1}^T \widetilde{a}_i^{k-1} \end{bmatrix}$$

and

$$\widetilde{b}^1 = b$$

$$\widetilde{b}^k = \begin{bmatrix} \widetilde{b}^{k-1} \\ u_{k-1}^T \widetilde{b}^{k-1} \end{bmatrix}$$

for $k = 2, \ldots, K$. In other words, $\widetilde{a}_i^k$ is the $i$th column of the constraint matrix of the IP and $\widetilde{b}^k$ is the constraint vector after applying cuts $u_1, \ldots, u_{k-1}$. An identical induction argument to that of Balcan et al. [10] shows that for each $k \in [K]$,

$$\left\lfloor u_k^T \widetilde{a}_i^k \right\rfloor \in \left[ -(1+U)^k \|a_i\|_1, (1+U)^k \|a_i\|_1 \right]$$

and

$$\left\lfloor u_k^T \widetilde{b}^k \right\rfloor \in \left[ -(1+U)^k \|b\|_1, (1+U)^k \|b\|_1 \right].$$

Now, as in the single-GMI-cut setting, consider the surfaces

$$\left\lfloor u_k^T \widetilde{a}_i^k \right\rfloor = \ell_i \iff \ell_i \leq u_k^T \widetilde{a}_i^k < \ell_i + 1 \tag{21}$$

and

$$\left\lfloor u_k^T \widetilde{b}^k \right\rfloor = \ell_0 \iff \ell_i \leq u_k^T \widetilde{b}^k < \ell_0 + 1 \tag{22}$$

for every $i, k$, and every integer $\ell_i \in [-(1+U)^k \|a_i\|_1, (1+U)^k \|a_i\|_1] \cap \mathbb{Z}$ and every integer $\ell_0 \in [-(1+U)^k \|b\|_1, (1+U)^k \|b\|_1] \cap \mathbb{Z}$. In addition, consider the surfaces

$$u_k^T \widetilde{a}_i^k - \ell_i = u_k^T \widetilde{b}^k - \ell_0 \tag{23}$$

for each $i, k, \ell_i, \ell_0$. As observed by Balcan et al. [10], $u_k^T \widetilde{a}_i^k$ is a polynomial in $u_1[1], \ldots, u_1[m], u_2[1], \ldots, u_2[m+1], \ldots, u_k[1], \ldots, u_k[m+k-1]$ of degree at most $k$ (as is $u_k^T \widetilde{b}^k$), so surfaces 21, 22, and 23 are all degree-$K$ polynomial hypersurfaces for all $i, k$. Within any connected component of $[-U, U]^m \times \cdots \times [-U, U]^{m+K-1}$ induced by these hypersurfaces, $\lfloor u_k^T \widetilde{a}_i^k \rfloor$ and $\lfloor u_k^T \widetilde{b}^k \rfloor$ are constant. Furthermore $\mathbf{1}[f_i^k \leq f_0^k]$ is invariant for every $i, k$, where $f_i^k = u_k^T \widetilde{a}_i^k - \lfloor u_k^T \widetilde{a}_i^k \rfloor$ and $f_0^k = u_k^T \widetilde{b}^k - \lfloor u_k^T \widetilde{b}^k \rfloor$.

Now, fix a connected component $D \subseteq [-U, U]^m \times \cdots \times [-U, U]^{m+K-1}$ induced by the above hypersurfaces, and let $C \subseteq \mathbb{R}^{K(n+1)}$ be the intersection of $q$ polynomial inequalities of degree at most $d$. Consider a single degree-$d$ polynomial inequality in $K(n+1)$ variables $y_1, \ldots, y_{K(n+1)}$, which can be written as

$$\sum_{\substack{T \subseteq [K(n+1)] \\ |T| \leq d}} \lambda_T \prod_{j \in T} y_j = \sum_{\substack{T_1, \ldots, T_K \subseteq [n+1] \\ |T_1| + \cdots + |T_K| \leq d}} \lambda_{T_1, \ldots, T_K} \prod_{j_1 \in T_1} y_{j_1} \cdots \prod_{j_K \in T_K} y_{j_K} \leq \gamma.$$

Now, the sets $S_1, \ldots, S_K$ defined by $S_k = \{i : f_i^k \leq f_0^k\}$ are fixed within $D$, so we can write this as

$$\sum_{\substack{T_1, \ldots, T_K \subseteq [n+1] \\ |T_1| + \cdots + |T_K| \leq d}} \lambda_{T_1, \ldots, T_K} \prod_{k=1}^K \left[ \prod_{\substack{j \in T_k \cap S_k \\ j \neq n+1}} f_j^k (1 - f_0^k) \prod_{\substack{j \in T_k \setminus S_k \\ j \neq n+1}} f_0^k (1 - f_j^k) \prod_{\substack{j \in T_k \\ j = n+1}} f_0^k (1 - f_0^k) \right] \leq \gamma.$$

We have that $f_j^k$ and $f_0^k$ are degree-$k$ polynomials in $u_1, \ldots, u_k$. Since the sum is over all multisets $T_1, \ldots, T_K$ such that $|T_1| + \cdots + |T_K| \leq d$, there are at most $d$ terms across the products, each of the form $f_j^k (1 - f_0)^k$, $f_0^k (1 - f_j^k)$, or $f_0^k (1 - f_0)^k$. Therefore, the left-hand-side is a polynomial of

degree at most $2dK$, and if $C \subseteq \mathbb{R}^{K(n+1)}$ is the intersection of $q$ polynomial inequalities each of degree at most $d$, the set

$$\{(\boldsymbol{u}_1, \ldots, \boldsymbol{u}_K) \in D : (\boldsymbol{\alpha}(\boldsymbol{u}_1, \ldots, \boldsymbol{u}_K), \beta(\boldsymbol{u}_1, \ldots, \boldsymbol{u}_K)) \in C\} \subseteq [-U, U]^m \times \cdots \times [-U, U]^{m+K-1}$$

can be expressed as the intersection of $q$ degree-$2dK$ polynomial inequalities.

To finish, we run this process for every connected component $C \subseteq \mathbb{R}^{K(n+1)}$ in the partition established by Theorem C.6. This partition consists of $O(12^n n^{2n} K^{2n^2} (m + 2n)^{2n^2} \tau^{5n^2})$ degree-$2K$ polynomials over $\mathbb{R}^{K(n+1)}$. By Warren's theorem and the Milnor-Thom theorem, these polynomials partition $\mathbb{R}^{K(n+1)}$ into $O(12^{n(n+1)} n^{2n(n+1)} K^{2n^2(n+1)} (m + 2n)^{2n^2(n+1)} \tau^{5n^2(n+1)})$ connected components. Running the above argument for each of these connected components of $\mathbb{R}^{K(n+1)}$ yields a total of $O\left(12^{n(n+1)} n^{2n(n+1)} K^{2n^2(n+1)} (m + 2n)^{2n^2(n+1)} \tau^{5n^2(n+1)}\right) \cdot$

$O\left(12^n n^{2n} K^{2n^2} (m + 2n)^{2n^2} \tau^{5n^2}\right) = 2^{O(n^2)} K^{O(n^3)} (m + 2n)^{O(n^3)} \tau^{O(n^3)}$ polynomials of degree $4K^2$. Finally, we count the surfaces of the form (21), (22), and (23). The total number of degree-$K$ polynomials of type 21 is at most $O(nK(1 + U)^K \|A\|_1)$, the total number of degree-$k$ polynomials of type 22 is $O(K(1 + U)^K \|\boldsymbol{b}\|_1)$, and the total number of degree-$K$ polynomials of type 23 is $O(nK(1 + U)^{2K} \|A\|_1 \|\boldsymbol{b}\|)$. Summing these counts yields the desired number of surfaces in the lemma statement.

In any connected component of $[-U, U]^m$ determined by these surfaces, $\mathbf{1}[(\boldsymbol{\alpha}(\boldsymbol{u}), \beta(\boldsymbol{u})) \in C]$ is invariant for every connected component $C \subseteq \mathbb{R}^{K(n+1)}$ in the partition of $\mathbb{R}^{K(n+1)}$ established in Theorem C.6. This means that the tree built by branch-and-cut is invariant, which concludes the proof. $\qquad \square$

Finally, applying the main result of Balcan et al. [9] to Lemma D.3, we get the following pseudo-dimension bound for the class of $K$ sequential GMI cuts at the root of the B&C tree.

**Theorem D.4.** *For $\boldsymbol{u}_1 \in [-U, U]^m, \boldsymbol{u}_2 \in [-U, U]^{m+1}, \ldots, \boldsymbol{u}_K \in [-U, U]^{m+K-1}$, let $g_{\boldsymbol{u}_1, \ldots, \boldsymbol{u}_K}(\boldsymbol{c}, A, \boldsymbol{b})$ denote the number of nodes in the tree B&C builds given the input $(\boldsymbol{c}, A, \boldsymbol{b})$ after sequentially applying the GMI cuts defined by $\boldsymbol{u}_1, \ldots, \boldsymbol{u}_K$ at the root. The pseudo-dimension of the set of functions $\{g_{\boldsymbol{u}_1, \ldots, \boldsymbol{u}_K} : (\boldsymbol{u}_1, \ldots, \boldsymbol{u}_K) \in [-U, U]^m \times \cdots \times [-U, U]^{m+K-1}\}$ on the domain of IPs with $\|A\|_1 \le a$ and $\|\boldsymbol{b}\|_1 \le b$ is*

$$O\left(mK^3 \log U + mn^3 K^2 \log(mnK\tau) + mK^2 \log(ab)\right).$$