# OpenReview forum: "Structural Analysis of Branch-and-Cut and the Learnability of Gomory Mixed Integer Cuts"
_NeurIPS.cc/2022/Conference — NeurIPS 2022 Accept_

### Official Review · Reviewer_zTZn · 2022-07-10

**Rating:** 8
**Confidence:** 4
**Soundness:** 3 good
**Presentation:** 3 good
**Contribution:** 4 excellent

**Summary:**

The paper develops a theoretical framework to investigate the geometric structure incurred by Gomory Mixed Integer (GMI) cuts and its implications for learning. More precisely, the authors first show the closed form of the optimal LP solutions obtained when adding a GMI cut, also demonstrating that the cut parameters lie in a countable number of connected degree-2 polynomial hypersurfaces.  Based on this result, they establish (surprising) conditions for which two or more sets of distinct GMI cuts would result in the same branch-and-cut tree. Finally, given a particular instance distribution, they leverage the previous structural results to derive the sample complexity associated with GMI, i.e., that the expected tree size over a training set of instances is sufficiently close to the expected average.

**Questions:**

1. Are all the cuts in a partition required? That is, can we obtain the same branch-and-cut trees if we omit some of them?

2. What would be the challenges in adding cuts throughout the search procedure, and not only at the root node? Would the sample complexity still be a valid bound?

**Strengths And Weaknesses:**

Strengths
- Very strong theoretical results, advance our understanding and provide further geometric insights into GMI cuts

Weaknesses
- Perhaps some of the results could be formalized more carefully
- There is little insight into how the theory translates into actual learning. It would be interesting to incorporate a discussion of the practical elements of the approach and the sample complexity

Major Comments

Overall, I believe the paper provides a very strong theoretical contribution to the extensive body of work associated with GMI cuts, which are key to the solution approach in modern mathematical programming solvers. In particular, I found the result stating that GMI cuts could be partitioned into sets with similar branch-and-cut trees quite surprising; in hindsight, that makes sense given the "discreteness" associated with the dual description of rational polyhedra. The paper also reads very well, with an interesting way of writing more on the informal side. However, that also brings a few concerns.

(i) [Rigour.] While I believe the results are sound, I felt that the proofs (or general statements) were written in a sometimes informal way and lacked a bit of discussion or explicit statements of boundary conditions. My apologies if I sound "picky" here, but for some illustrative examples...

(a) the authors state multiple times that, when adding a cut, the optimum would "jump" or "change" from one vertex to another in the LP; however, adding a cut changes the polytope itself, so it is not clear what "LP" this refers to.

(b) I am sure results still hold, but what if the LP does not have any extreme points  (e.g., empty or unbounded); I felt it is missing some assumptions (e.g., full rank of A?) to ensure this would not happen (unless I missed something obvious?). Theorem 3.1 is easy to understand (essentially a nice application of Cramer's rule), but the description was a bit hand-wavey; would it be more formal to state that the proof essentially enumerates the basis of the LP considering that the cut is part of the basis or not, as opposed to just using the geometric intuition in the argument?

(c) In the proof of Lemma 4.1: "we assume that the B&C tree takes some special form denoting an invalid cut" - my apologies as this was a bit unclear to me, it would be great if the authors could clarify...

(ii) [Learning.] Until I reached Section 5, I was under the impression that I was reading a (very high-quality) paper in the Mathematical Programming journal. The learning element (here implied by the sample complexity theory) seems secondary because there is little intuition about its practical aspects. There are some hints in the conclusion that the theory opens doors for new cut generation mechanisms, which I agree with, but how does that actually impact learning the parameters? It seems that the motivation of the Intro and the actual results are somewhat disjoint. I would suggest authors add more comments on what Theorem 5.3 means when learning parameters (e.g., when would "too many samples" be needed), considering the broader audience of the conference.


Minor notes
- Text sometimes alternates between "piecewise" and "piece-wise"

---

> ### Author Response · Authors · 2022-08-02
> **Response to Reviewer zTZn**
>
> Thank you for the review! We respond to your specific questions and comments below.
>
> > “While I believe the results are sound, I felt that the proofs (or general statements) were written in a sometimes informal way and lacked a bit of discussion or explicit statements of boundary conditions.”
>
> We believe all our proofs are fully rigorous: any argument that includes geometric intuition is backed up by appeal to facts about systems of linear equations. We have aimed to make our technical contributions as readable and accessible as possible to the broader NeurIPS audience, and therefore we included the corresponding geometric intuition whenever possible. To make things even more clear, we will update Section 2 to include the formal definitions of edges and vertices, with reference to the excellent chapter on polyhedral theory in the textbook “Integer Programming” by Conforti, Cornuéjols, and Zambelli.
>
> We agree that we should have been more explicit in elaborating on some of the boundary conditions (or assumptions that rule out boundary conditions). We address the specific ones you raised below.
>
> >“The authors state multiple times that, when adding a cut, the optimum would ‘jump’ or ‘change’ from one vertex to another in the LP; however, adding a cut changes the polytope itself.”
>
> Our goal was to convey intuitively that a cut intersects the original LP polytope P = {x : Ax <= b, x >= 0} at some number of edges of P. The new LP optimum is achieved at a vertex that lies on one of these edges. As the cut varies, the edge of P on which the new LP optimum is achieved might change (this is what we meant by the optimum jumping from one vertex/edge to the other). We will make this clearer.
>
> >“What if the LP does not have any extreme points (e.g., empty or unbounded)”
>
> We assume that the polytope P = {x: Ax <= b, x >= 0} is full dimensional, that is, dim P = n. This is just for expositional ease, all arguments go through whatever dim P is. We will state this clearly. We assume in Section 2 that P is bounded (so the LP relaxation of the input IP is bounded). Our results actually still apply even if P is unbounded (assuming the LP relaxation has a bounded optimum). This is due to the Minkowski-Weyl theorem which states that any rational polyhedron can be decomposed as the Minkowski sum of a rational polytope (bounded) and its recession cone. The analysis in our paper can be run on this rational polytope. If P is nonempty but contains no integer feasible points, that is OK and our analysis handles that. Finally, if P itself is empty, our analysis handles this vacuous case (all cuts behave identically, and branch-and-cut terminates at the root).
>
> > Theorem 3.1: “Would it be more formal to state that the proof essentially enumerates the basis of the LP considering that the cut is part of the basis or not, as opposed to just using the geometric intuition in the argument?”
>
> As mentioned above, we will include formal definitions of edges and vertices, and include a reference to the chapter on polyhedral theory in the textbook by Conforti, Cornuéjols, and Zambelli, to make this clearer. Given these formalisms, we strongly believe that the way our argument is written is as rigorous and formal as one that considers the optimal LP basis more explicitly.
>
> > What does it mean for the B&C tree to “some special form denoting an invalid cut”?
>
> We mean, for example, that the B&C algorithm outputs ERROR if an invalid cut is added to the IP. Then, in the partition of space established by our main structural result for B&C, there are connected components consisting of invalid cuts that all yield ERROR instead of a valid B&C search tree. Now, when we analyze the coefficient map defined by GMI cuts, such regions will never be considered since GMI cuts are by definition valid. This assumption of outputting ERROR for invalid cuts is stylistic – if an invalid cut is added to the IP formulation, B&C is not going to be able to detect invalidity and will still execute and build a tree. Our structural result nevertheless still applies, and we will simply have regions where cuts from that region are invalid but yield the same B&C tree. These “erroneous” regions, as mentioned, will never be touched by the GMI coefficient map. We will elaborate on this in the appendix.
>
> >“Are all the cuts in a partition required? That is, can we obtain the same branch-and-cut trees if we omit some of them?”
>
> Indeed, it is possible that some of the parameterized GMI cuts are redundant, and thus the same B&C tree could be built if they were omitted.
>
> >Adding cuts throughout the tree.
>
> This is a very interesting and challenging direction for future research that we will add to Section 6.

---

> > ### Comment · Reviewer_zTZn · 2022-08-09
> > **Feedback**
> >
> > Thank you for your reply and thorough response. I understand and agree with your point of providing some geometric intuition. Perhaps my concern is that I felt some sentences were vague in the proofs and paper compared to more theoretical & fundamental Math Prog. works (the example I provided, "with less of generality," or "bonding aspect is suppressed" and others). I would be satisfied with adding more precise/algebraic descriptions as the authors suggested in the response (my training was also based on Conforti et al), so thank you. Note that I am not mentioning the counting arguments, which were great.
> >
> > Finally, it was a bit disappointing that no comment was made concerning the "learning" aspect mentioned in my review; I still feel that this paper is much broader/more impactful and that the "learning" aspect reads as a minor portion of it.

---

### Official Review · Reviewer_JLsy · 2022-07-11

**Rating:** 8
**Confidence:** 4
**Soundness:** 3 good
**Presentation:** 4 excellent
**Contribution:** 3 good

**Summary:**

The authors analyze the sample complexity bounds required for learning to select from the infinite family of GMI cutting planes. The authors demonstrate that there are finitely many distinct outcomes of the branch and cut procedure by determining how the cuts impact the result at any branch and bound node, determine when the branch and cut algorithm results in the same result due to branching, node fathoming, and integral optima at intermediate nodes. After bounding this uniqueness, the authors determine the sample complexity for learning to select GMI cuts. Overall, this work builds on previous work of sample complexity in branch and bound which is mainly concerned with bounding the number of distinct predictive algorithms by analyzing the impact of different decisions in the branch and bound solver. Generally, these approaches rely on the same principle of bounding the number of distinct outcomes that the algorithm can have and then drawing conclusions of sample complexity on top of this bound.

**Questions:**

How easy is it to generalize this method to other types of cuts, or even other types of cut-based branching such as multi-variable branching?

Is it possible to generalize these results to settings like bilevel optimization or solving Stackelberg games?


**Limitations:**

The authors adequately address the limitations of their approach and any ethical considerations

**Strengths And Weaknesses:**

The main strength is that this work theoretically analyzes the requirements for any algorithm to determine a high-quality gutting plane procedure. The approach itself is interesting and tackles a new component of branch and bound. The approach additionally paves the way for future work on other aspects of optimization solvers that rely on selecting from an infinite family of objects as previous work was concerned with decisions such as variable selection where only a finite number of decisions are possible. Additionally, the paper is very well and clearly written and would be a good contribution to the area of theoretically analyzing learning algorithms meant to improve optimization procedures.

The main area for improvement would be to provide a general framework for other cutting plane algorithms such as where there are many latent constraints that are iteratively generated. Such situations arise in settings like bilevel optimization or game theory, where the solving process iteratively generates solutions to the internal level or adversary strategy respectively to compute the top level solution based on the performance with respect to the current known second level solution set. These second level solutions generally become cuts in the overall problem. Another setting which would be interesting for analysis is in determining a methodology for helping understand unknown categories of cuts that have yet to be devised. In these settings, a novel cut family is proposed which helps the reformulated problem solve much faster. It would be helpful to know how much benefit is possible from devising the ideal family of cuts for a given problem.

---

> ### Author Response · Authors · 2022-08-02
> **Response to Reviewer JLsy**
>
> Thank you for the review! Also, thank you for the suggestions of very interesting directions for future research (namely, generalizing our results to other settings like bilevel optimization, solving Stackelberg games, and multi-variable branching). If accepted, we will use the extra page in the camera-ready version to include these future directions. Moreover, we expect that our results could also be used to analyze other cut families and that Section 5 could provide guidance for doing so.

---

### Official Review · Reviewer_H9iB · 2022-07-12

**Rating:** 8
**Confidence:** 2
**Soundness:** 4 excellent
**Presentation:** 4 excellent
**Contribution:** 4 excellent

**Summary:**

The paper proves a number of results regarding the sample complexity
of gomory mixed integer cuts.


**Questions:**

None.

**Limitations:**

It would help if the paper included some experiments.

**Strengths And Weaknesses:**

Strengths:
A number of interesting results are proved, some of which do not seem
expected.

Weaknesses:
Several key aspects of the paper are left for the appendix, and in
total the submission in 30 pages. Also, the lack of experiments in the
main text is problematic.

---

> ### Author Response · Authors · 2022-08-02
> **Response to Reviewer H9iB**
>
> Thank you for the review! We are glad you found the results surprising and agree that empirical results would be a great direction for future research.

---

### Official Review · Reviewer_3NW3 · 2022-07-12

**Rating:** 8
**Confidence:** 3
**Soundness:** 4 excellent
**Presentation:** 3 good
**Contribution:** 4 excellent

**Summary:**

This paper applies learning theory to provide sample complexity bounds for GMI cuts.

The paper is very dense, and the literature is updated and exhaustive.

The main results of this paper are the three lemmas and the theorem presented in Section 4, which permit to the author to express their sample complexity bounds for GMI cuts given in Section 5. The authors convey the main idea in the introduction, given the main background on B&C in Section 2, and then the results are very interesting, and the proofs are very technical.

For a reader who is not an expert of this topic, it is difficult to follow the paper without consulting the material in the appendix (which is 15 pages long).


**Questions:**

- How do you think we could exploit your results to implement more efficient B&C algorithms based on GMI? Could you expand the final comments on your conclusions?



**Limitations:**

We see no limitations in this paper.

**Strengths And Weaknesses:**

STRENGTHS: the topic of this paper is very technical and hard to follow without a strong theoretical background, but we think this paper is laying the foundation of learnability principles for branch-and-cut algorithms in general, and for Gomory Mixed Integer cuts, specifically.

WEAKNESS: We think the paper suffers the 9-page limit imposed by the conference venue. Given that the appendix, which is very useful for understanding the contributions, is another 15 pages long, the paper should be sent directly to a journal (e.g., the new "Transactions on Machine Learning Research", https://www.jmlr.org/tmlr/).

---

> ### Author Response · Authors · 2022-08-02
> **Response to Reviewer 3NW3**
>
> Thank you for the review! We respond to your question below.
>
> > “How do you think we could exploit your results to implement more efficient B&C algorithms based on GMI? Could you expand the final comments on your conclusions?”
>
> In future research, the piecewise partition that we uncover might be useful for designing efficient algorithms to learn the optimal GMI cut parameters for the distribution. (This type of structure was useful for designing learning algorithms in a very different context by Balcan et al. [ICLR’20], for example.) This could suggest new, data-driven ways of generating GMI cuts beyond those implied by the simplex tableau.

---

### Meta-Review · Area_Chair_LDtV · 2022-08-21

**Recommendation:** Accept
**Confidence:** Certain

**Metareview:**

There is general agreement that this is a strong paper that should be accepted. I agree with the discussion. Nothing much to add.

**Award:**

No

---

### Decision · Program_Chairs · 2022-09-14

Accept